# An NLR paralog Pit2 generated from tandem duplication of *Pit1* fine-tunes Pit1 localization and function

Yuying Li[1,2,19], Qiong Wang [2,3,19], Huimin Jia [4,19], Kazuya Ishikawa[2,5], Ken-ichi Kosami[2,6], Takahiro Ueba[7], Atsumi Tsujimoto[7], Miki Yamanaka[7], Yasuyuki Yabumoto[7], Daisuke Miki [2], Eriko Sasaki [8], Yoichiro Fukao[9], Masayuki Fujiwara[10], Takako Kaneko-Kawano [11], Li Tan[2], Chojiro Kojima [12], Rod A. Wing[13], Alfino Sebastian[14], Hideki Nishimura[14], Fumi Fukada [14], Qingfeng Niu [15], Motoki Shimizu[16], Kentaro Yoshida[17], Ryohei Terauchi[16,17], Ko Shimamoto[7] & Yoji Kawano [2,14,18] ✉

NLR family proteins act as intracellular receptors. Gene duplication amplifies the number of NLR genes, and subsequent mutations occasionally provide modifications to the second gene that benefits immunity. However, evolutionary processes after gene duplication and functional relationships between duplicated NLRs remain largely unclear. Here, we report that the rice NLR protein Pit1 is associated with its paralogue Pit2. The two are required for the resistance to rice blast fungus but have different functions: Pit1 induces cell death, while Pit2 competitively suppresses Pit1-mediated cell death. During evolution, the suppression of Pit1 by Pit2 was probably generated through positive selection on two fate-determining residues in the NB-ARC domain of Pit2, which account for functional differences between Pit1 and Pit2. Consequently, Pit2 lost its plasma membrane localization but acquired a new function to interfere with Pit1 in the cytosol. These findings illuminate the evolutionary trajectory of tandemly duplicated NLR genes after gene duplication.

Nucleotide-binding domain and leucine-rich repeat (NLR) family proteins serve as intracellular immune receptors that perceive effector proteins secreted by pathogens and thus induce immune responses termed effector-triggered immunity (ETI)[1–5]. NLR proteins consist of a tripartite domain architecture: an N-terminal region, a central NB-ARC (nucleotide-binding and APAF-1, certain resistance gene products and CED-4) domain, and C-terminal leucine-rich repeats (LRRs). Their N-terminal structures include either a coiled-coil (CC) domain or a Toll-interleukin 1 receptor (TIR) domain[1]. Functionally, the N-terminal CC and TIR domains mainly contribute to the perception of pathogen effectors and signal transduction; the NB-ARC domain has ATP binding and hydrolysis activities to serve as a switch domain; and the LRR domain is a key determinant for pathogen effector recognition and

self-regulation[1,6–8]. The recent three-dimensional analyses have revealed how both CC and TIR type NLRs form a pentameric or tetrameric resistosome to trigger cell death and immunity[9–14].

NLR proteins are one of the most expanded and diversified protein families in plants[2,15,16]. Whole genome, segmental, and gene duplication contribute to the increment of NLR genes[17–21], which produces multiple layers of complexity consisting of genetically linked NLR pairs and NLR networks[22–24]. Typically, single NLRs are self-sufficient in perceiving pathogen effectors and triggering immune responses (called singleton NLR). In the cases of NLR pairs and networks, during evolution, NLRs specialize their roles in two functions: serving as sensors that recognize pathogen effectors or as helpers (also known as executors) that induce immune responses[25–27]. One of

the cooperatives is sensor–helper NLR pairs, wherein two NLR proteins work together to regulate ETI. Some NLR pairs, such as *Arabidopsis* RPS4-RRS1 and rice RGA4-RGA5, physically form heterocomplexes in which the two NLRs have antagonistic functions[28–30]. Helper NLRs induce immune responses while sensor NLRs possess two functions: recognition of pathogen effectors and suppression of helper NLRs. In the other cooperative mechanisms, NLR signaling is mediated by helper NLRs, which serve as downstream signaling hubs for a diverse array of sensor NLRs[25,31]. Sensor NLRs have undergone massive expansion and diversification, but helper NLRs have limited diversification and redundancy. Sensor and helper NLRs collectively form NLR networks[31]. A subset of sensor NLRs have atypical integrated domains (IDs), which are derived from pathogen-host targets that recognize the pathogen effectors[32,33].

The function of plant NLRs in their breath of recognition spectra is believed to have evolved through interaction with fast-evolving pathogens. Tandemly aligned NLRs are widespread in plant genomes[34,35], and analyzing these NLRs provides insights into the evolution of NLRs. The sensor-executor NLR pair RPS4/RRS1 and its duplicate pair RPS4B/RRS1B exhibit different specificity to effectors[36]. The RRS1-R/RPS4 pair recognizes two bacterial effectors: *Pseudomonas syringae* AvrRps4 and *Ralstonia solanacearum* PopP2, but the RRS1B/RPS4B pair recognizes only AvrRps4. Another sensor-executor NLR pair of *Arabidopsis* CHS3/CSA1 divides into three clades: NLR-ID (clade 1) and NLR-non-ID alleles (clade 2/3)[37]. An ancestral *Arabidopsis* CHS3/CSA1 pair seems to have acquired a new recognition specificity and activation mechanisms with ID acquisition. Both ID and non-ID alleles of the CHS3/CSA1 pair are retained in various *Arabidopsis* populations, suggesting balancing selection. Interestingly, artificial combinations [CHS3(clade 1)/CSA1(clade 2/3), CHS3(clade 2/3)/CSA1(clade 1), RRS1/RPS4B, or RRS1B/RPS4] are not functional[36,37]. Moreover, authentic alleles of rice Pik-1/Pik-2 pairs trigger effective immune responses, while mismatched pairs lead to autoimmune phenotypes[38]. A single amino acid polymorphism between matching Pik NLR pairs largely explains this allelic specialization. NLRs are strictly regulated to mount effective and regulated immune responses against pathogens. Mismatched NLRs, which can cause deleterious autoimmune phenotypes known as Dangerous Mix phenotypes, may be eliminated from populations[39,40]. NLR-non-ID genes are more extensive than NLR-ID genes in plant genomes[41], but most of the previous works on tandemly aligned NLRs have investigated the NLR-ID type. Therefore, the evolutionary trajectory of NLR-non-ID receptor pairs remains largely unknown.

NLR proteins require proper subcellular localization to trigger their various immune responses[1]. Many NLR proteins show nucleo-cytoplasmic distribution, among which MLA10, Pib, and N play key roles in immunity by interacting with transcription factors in the nucleus[42–44]. Some NLR proteins such as RPS5, RPS2, and RPM1 anchor themselves either directly or indirectly to the plasma membrane through lipidation or by binding to a plasma membrane-localized guardee protein known as RIN4[45–47]. The NLR protein RPS5 is reported to localize at the plasma membrane through two lipid modifications, namely myristoylation and palmitoylation[45]. The *Arabidopsis* NLR protein ZAR1 forms a pentameric complex named resistosome upon activation[12,14], which serves as a calcium-permeable channel at the plasma membrane to trigger immunity and cell death[13]. Although our knowledge of NLR protein localization is accumulating, the relationship between NLR localization and immune function remains elusive.

The NLR-type *resistance* (R) gene *Pit* (hereafter called *Pit1*) was identified in an Indonesian rice variety, Tjahaja, and shows broad-spectrum resistance to various races of the rice blast fungus *Magnaporthe oryzae*[48,49]. We have previously revealed that the rice small GTPase OsRac1 acts as an important molecular switch, working downstream of Pit1[50,51]. Overexpression of a constitutively active form

of Pit1 triggers cell death in *Nicotiana benthamiana*, and co-expression of the dominant negative form of OsRac1 attenuates this cell death. Palmitoylation is essential for plasma membrane localization of Pit1 and is involved in its interaction with and activation of OsRac1[51]. We recently found that a GDP/GTP exchange factor (GEF) protein, OsSPK1, is a direct downstream target of Pit1, and functions as an activator for OsRac1 to induce disease resistance to rice blast fungus[52]. However, the mode of action of Pit1 in relation to other NLRs is not known.

In this study, we tried to identify Pit1 binding partners and isolated Pit1's paralogue Pit2 as a Pit1 binding protein. Pit2 formed a stable heterocomplex with Pit1 and suppressed Pit1-triggered cell death. Two important residues in the NB-ARC domain cause the differences in function and localization between Pit1 and Pit2. Moreover, our analysis revealed how the *Pit1* and *Pit2* genes evolved from an ancestral *Pit* gene. These findings considerably extend our understanding of the regulation mechanism of pairs of NLR proteins, and shed light on the evolution of duplicated NLR genes.

## Results

### NLR proteins Pit1 and Pit2 form heteromers

Pit is an NLR-type R protein and plays a critical role in disease resistance to rice blast fungus[49]. To illuminate how Pit triggers ETI, we tried to identify proteins that interact with Pit and generated rice suspension cells expressing a constitutively active form of Pit fused with FLAG tag (Pit D485V-FLAG) driven by an estradiol-inducible promoter. We could detect the expression of Pit D485V at 8 and 30 h after estradiol treatment and performed an immunoprecipitation assay at these time points (Supplementary Figs. 1A and 3B). Interestingly, when Pit D485V-FLAG was precipitated with anti-FLAG antibody, we identified six NLR proteins and two small GTPases, OsRac1 and OsRac2, in the precipitates (Supplementary Table 1). Hereafter, we rename the original Pit (AB379816 and LOC_Os01g05620) as Pit1 and LOC_Os01g05600.1 as Pit2, and the remaining five NLR proteins in the Pit1 D485V precipitate as Pit1-associated NLR1-5 (PAN1–PAN5)[49]. The exponentially modified protein abundance index (emPAI) shows protein abundance from peptide counts in a single LC-MS/MS experiment and the emPAI score for Pit2 (AB379819 and LOC_Os01g05600) was the highest among the six NLR proteins, indicating that Pit2 is the most abundant NLR protein in the precipitate with Pit1 D485V. Moreover, the *Pit2* gene is located adjacent to the *Pit1* gene, and the distance between them is 9 kbp (Supplementary Fig. 1B). Pit1 and Pit2 show 88% amino acid identity, implying that *Pit1* and *Pit2* were tandemly duplicated from an ancestral *Pit* gene and are functionally linked[49] (Supplementary Fig. 3A). However, the amino acid identity between Pit1 and PAN1-5 is no more than 34% (Supplementary Table 1 and Supplementary Fig. 2) and, therefore, we focused on *Pit2* for further analyses.

Recent evidence indicates that sensor–helper (also called executor) NLR pairs such as RGA4–RGA5 and RRS1–RPS4 directly associate with each other and work together to sense pathogens and induce immune responses[28,29,53]. These results raise the possibility that Pit2 forms a complex with Pit1. To test this, co-immunoprecipitation (co-IP) assays were performed by transiently expressing Pit1 with Pit2 in *N. benthamiana* leaves. When HA-tagged Pit1 was precipitated by anti-HA antibody, Pit2-Myc but not another NLR protein RGA4-Myc coprecipitated with Pit1-HA, demonstrating Pit2 forms heteromers with Pit1 (Fig. 1A). The result of this co-IP in *N. benthamiana* was consistent with that in rice protoplasts (Supplementary Fig. 1C). Moreover, we generated transgenic rice plants carrying Pit1-Flag and Pit2-Myc driven by their native promoters, and performed co-IP assays. When Pit2-Myc was precipitated by anti-Myc antibody, Pit1-Flag coprecipitated with Pit2-Myc (Supplementary Fig. 1D).

It has been reported that the N-terminal CC or TIR domain of NLRs is important for the heterooligomer interaction and immune signal transduction[28,30,54]. To further examine intermolecular interactions between Pit1 and Pit2, we performed a yeast two-hybrid assay using Pit

CC (Pit1: amino acids 1–140; Pit2: 1–140) domains. The CC domain of Pit2 forms heteromers with that of Pit1 in a yeast two-hybrid assay (Supplementary Fig. 1E).

## Pit2 suppresses Pit1-mediated cell death

We have previously reported that the overexpression of a constitutively active MHD motif mutant of Pit1 (Pit1 D485V) induces cell death through the activation of small GTPase OsRac1 homolog(s) in *N. benthamiana* without pathogen effector[50]. To examine whether Pit2 also has similar activity, we generated and transiently expressed the corresponding MHD motif mutant of Pit2 in *N. benthamiana* (Pit2 D484V). Under the same conditions in which Pit1 D485V strongly and Pit1 WT slightly elicited cell death, there were no discernible effects in leaves expressing either Pit2 WT or Pit2 D484V (Fig. 1B). To monitor whether Pit2 D484V activates OsRac1, which is a crucial switch molecule in Pit1 signaling in vivo, we employed a Förster resonance energy transfer (FRET) sensor called Ras and interacting protein chimeric unit (Raichu)-OsRac1[50,55]. This sensor can estimate activation levels of OsRac1 based on the intramolecular interaction between the active GTP-OsRac1 and the Cdc42/Rac interactive binding (CRIB) domain. When OsRac1 is activated, the active GTP-bound form of OsRac1 binds to the CRIB domain of PAK and this interaction brings CFP closer to Venus, leading to FRET from CFP to Venus (Supplementary Fig. 4A). Thus, the ratio of Venus/CFP fluorescence indicates the activation level of OsRac1 in vivo. The emission ratio of Venus/CFP in protoplasts expressing Pit2 D484V was much lower than that in protoplasts expressing Pit1 D485V (Fig. 1C), indicating that Pit2 D484V lacks the ability to activate OsRac1 in rice protoplasts. Western showed that all Pit mutant proteins fused HA tag were successfully expressed in *N. benthamiana* leaves but that the expression level of Pit2 D484V was much higher than that of Pit1 D485V (Supplementary Fig. 4B). Taken together, these results indicate that Pit2 D484V is unable to induce cell death and OsRac1 activation because of its nature and not because of its low protein expression.

Since Pit1 formed heteromers with Pit2, we co-expressed Pit1 with Pit2 in *N. benthamiana* to observe immune responses and found that the expression of Pit1 WT with control GUS elicited cell death, but that this was suppressed by the co-expression of Pit2 (Fig. 1D and Supplementary Fig. 4C), and Pit2 could not attenuate RGA4-mediated cell death. To confirm this result in a rice system, we deployed a luciferase reporter system that measures the viability of rice protoplasts based on luminescence using *Oryza sativa indica* Oc cells. Consistent with the results in *N. benthamiana*, luciferase activity was low in protoplasts expressing Pit1, showing that Pit1-triggered cell death in rice protoplasts. Pit2 could not induce cell death but was able to suppress Pit1-triggered cell death in rice protoplasts (Fig. 1E and Supplementary Fig. 4D). Taken together, these results suggest that Pit2 forms heteromers with Pit1 to suppress Pit1-mediated cell death. To test whether endogenous Pit2 affects Pit1 activity, we transiently silenced the *Pit2* gene by RNA interference (RNAi) in protoplasts from rice cultivar K59, which carries functional *Pit1* and *Pit2* genes, to monitor cell death activity[49], and revealed that moderate cell death occurred in *Pit2* RNAi protoplasts (Fig. 1F and Supplementary Fig. 4E). We examined whether endogenous *Pit1* is involved in cell death induced by *Pit2* RNAi, by performing the same assay using *Pit1* knockout (KO) K59 suspension cells. In the *Pit1* KO background, there was no obvious effect of *Pit2* RNAi, indicating that endogenous Pit1 mediates *Pit2* RNAi-induced cell death (Fig. 1F and Supplementary Fig. 4E). Interestingly, the transient knockdown of *Pit2* enhanced the expression of *PAN3* in K59 rice suspension cells, implying that *Pit2* is genetically linked to *PAN3* and PAN3 also contributes to Pit1-mediated cell death (Supplementary Fig. 4F). To investigate this possibility, we conducted the cell death assay using the *PAN3* knockdown K59 suspension cells. The transient knockdown of *PAN3* resulted in cell death mediated by Pit1, suggesting that PAN3 also functions as a suppressor to Pit1-induced cell death (Supplementary Fig. 4G).

To further test whether Pit2 contributes to Pit1-mediated disease resistance to rice blast fungus, we generated *Pit1* and *Pit2* knockout (KO) plants using CRISPR/Cas9 in the rice cultivar K59 which carries functional *Pit1* and *Pit2* genes[49]. We tried to generate *Pit2* KO twice; in the first trial, we did not obtain any plants but acquired 6 independent *Pit2* KO plants showing one allele during the second trial (Supplementary Fig. 5A, B). *Pit1* is not functional due to its low expression in the rice cultivar Nipponbare (Nip). As previously reported, *Pit1* KO compromised disease resistance to avirulent rice blast fungus race 007.0, and displayed enhanced fungal growth[49] (Fig. 1G, H and Supplementary Fig. 5C). *Pit2* KO also developed slightly larger lesions induced by avirulent rice blast fungus compared to K59 WT, suggesting that Pit2 also participates in resistance to rice blast fungus, but the effect of *Pit2* KO is much weaker than that of *Pit1* KO. In addition, the lesion induced by the virulent rice blast fungus (IB14-1K-1) in the *Pit1* KO and *Pit2* KO mutants was comparable to that in K59 WT (Supplementary Fig. 5D, E).

## Pit2 competes with Pit1 to form stable heteromers

The N-terminal CC domains of NLR proteins MLA10 and Sr33 contribute to their own homooligomerization and the consequent induction of immune responses[3,54]. We therefore performed co-IP assays in *N. benthamiana* to test whether Pit1 and Pit2 also form homocomplexes. Full-length Pit1 and Pit2 formed homomeric associations (Fig. 2A and Supplementary Fig. 6A). Pit1 and Pit2 form three different complexes: the two homomers (Pit1-Pit1 and Pit2-Pit2) (Fig. 2A) and the Pit1-Pit2 heteromers (Supplementary Fig. 1C). We further detected these three complexes using the CC domains of recombinant Pit1 and Pit2 proteins in an in vitro GST pull-down assay. Interestingly, the heteromerization of Pit1 and Pit2 was much stronger than the homomeric association of Pit1 (Fig. 2B), suggesting that Pit2 competitively attenuates Pit1 homocomplex formation by forming heteromers with Pit1. To test this hypothesis, we compared the Pit1 homocomplex in the presence and absence of Pit2 in *N. benthamiana* and rice protoplasts and found that in the presence of Pit2, Pit2 interacted with Pit1 and, concomitantly, the Pit1 homocomplex decreased (Fig. 2C and Supplementary Fig. 6B, C). An in vitro competitive binding assay also showed that the Pit1 homocomplex declined as the amount of Pit2 increased (Fig. 2D). Collectively, these data suggest that Pit1 forms a heterocomplex with Pit2 that is more stable than the Pit1 homomer.

## Pit2 captures Pit1 in the cytosol

We have previously reported that Pit1 localizes on the plasma membrane through its lipid modification palmitoylation[51] (Fig. 2E). To investigate whether Pit2 shows a similar subcellular distribution, we expressed the fluorescent protein Venus fused to Pit2 (Pit2-Venus) in rice protoplasts. Unlike Pit1-Venus, Pit2-Venus did not accumulate at the plasma membrane (Fig. 2E). Pit2-Venus hardly merged with mCerulean fused to a nuclear import signal but strongly colocalized with mCherry outside the nucleus, indicating that Pit2-Venus localizes mainly in the cytosol (Fig. 2E and Supplementary Fig. 6D). Next, to check the localization of Pit1 in the presence of Pit2, we co-expressed Pit1-Venus with Pit2-mCherry in rice protoplasts. In the presence of Pit2, the plasma membrane association of Pit1 was dramatically reduced and most Pit1 accumulated in the cytosol where Pit2 was also enriched (Fig. 2F). To confirm this localization pattern, we employed biochemical fractionation by transiently co-expressing Pit1-HA and Pit2-Myc in *N. benthamiana*. Pit1-HA alone accumulated mainly in the microsomal fraction, whereas Pit2-Myc alone was predominantly in the soluble fraction (Fig. 2G and Supplementary Fig. 6E). However, in the presence of Pit2-Myc, microsomal Pit1-HA decreased and, concomitantly, soluble Pit1-HA increased. These results imply that Pit2 forms a heterocomplex with Pit1 in the cytosol.

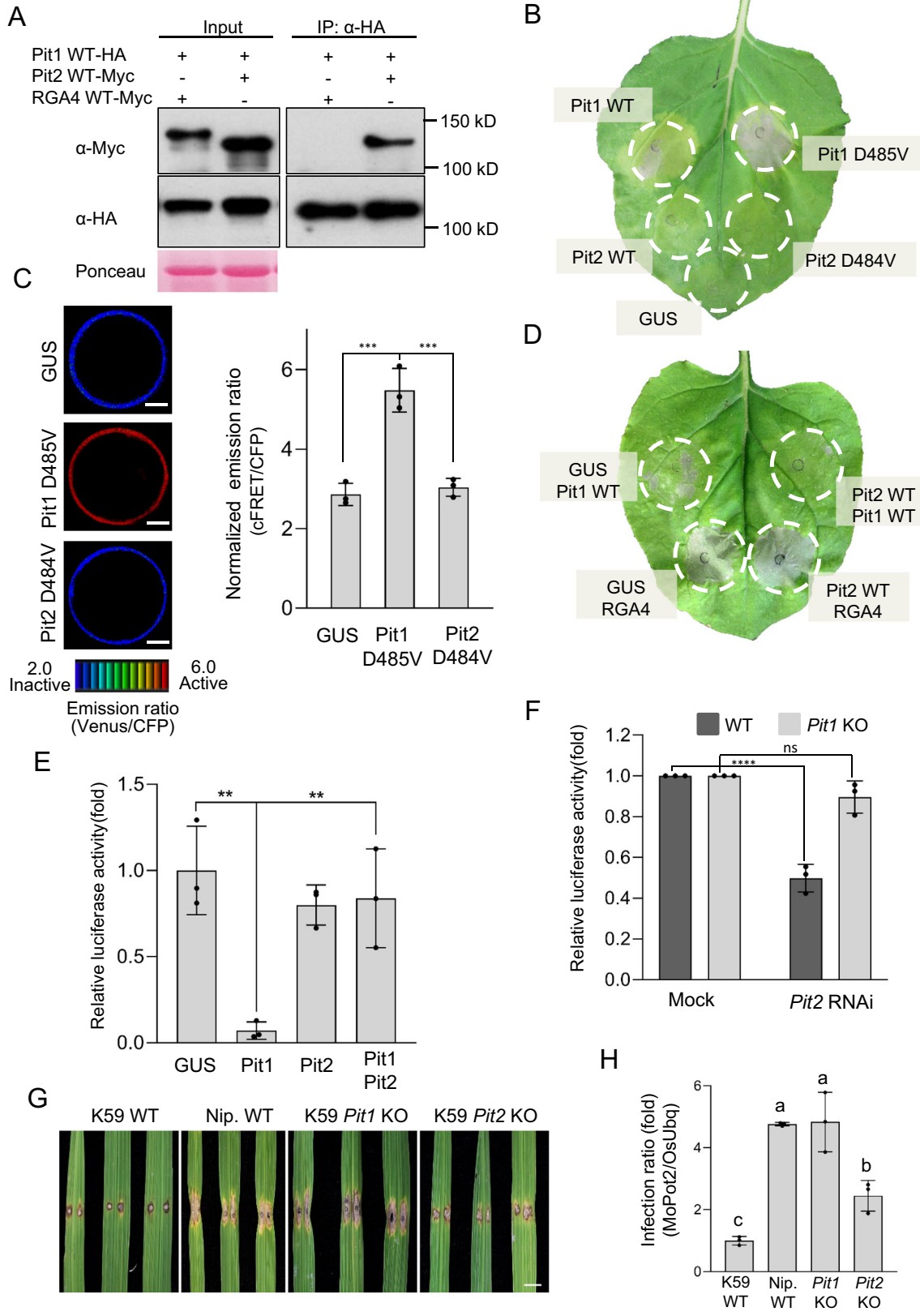

## Three residues in the NB-ARC domain determine the functional differences between Pit1 and Pit2

It is noteworthy that Pit1 and Pit2 share 88% protein identity but possess different functions. We next tried to decipher the molecular mechanism underlying these differences. We have previously identified two important direct interactors of Pit1, namely the small GTPase OsRac1 and its activator OsSPK1, which play important roles in Pit1-mediated resistance to rice blast fungus[50–52]. We first compared the binding activity to OsRac1 and OsSPK1 between Pit1 and Pit2 by co-IP assays using *N. benthamiana*, and found that the binding activity of Pit2 to OsRac1 and OsSPK1 is comparable to that of Pit1 (Supplementary Fig. 7A, B), implying that some other

**Fig. 1 | Pit2 interacts with Pit1 *in planta* and suppresses Pit1-mediated cell death. A** In vivo interaction of full-length Pit1 and Pit2. The indicated tagged proteins were transiently expressed in *N. benthamiana*. Co-IP was performed and proteins were detected by Western blot. Ponceau staining of Rubisco serves as a loading control. Data was repeated three times with similar results. **B** The indicated proteins were transiently expressed in *N. benthamiana*. GUS serves as a negative control. Cell death was photographed at 3 dpi. **C** Activation of OsRac1 by Pit1 and Pit2 using Raichu-OsRac1 FRET in vivo. Color scale of emission ratio images (left) in rice protoplasts co-expressing Raichu-OsRac1 represents the level of OsRac1 activation. Scale bars, 5 μm. The bar graph (right) indicates statistical analysis of OsRac1 activation. Bars = mean ± s.d. (*n* = 3 biological replicates). **D** The indicated combinations of proteins were transiently expressed in *N. benthamiana* leaves. Cell death was photographed at 3 dpi. **E** Cell death activity of indicated proteins in rice protoplasts. The indicated constructs were co-transfected with a *luciferase* reporter

vector. *Luciferase* activity was measured 40 h after transfection. Relative luciferase activity (GUS = 1) is shown. Bars = mean ± s.d. (*n* = 3 biological replicates). **F** Rice protoplasts from K59 WT or *Pit1* KO were co-transfected with RNAi constructs against *Pit2* and the *luciferase* reporter vector. *Luciferase* activity was measured 40 h after transfection. Relative luciferase activity (Mock = 1) is shown. Bars = mean ± s.d. (*n* = 3 biological replicates). The asterisks indicate significant differences determined by one-way (**C** and **E** with Tukey's test) or two-way (**F** with Šídák's test) ANOVA. $**P < 0.01$, $***P < 0.001$, $****P < 0.0001$, ns indicates no significant difference. **G** Infection assays of *Pit1* and *Pit2* KO plants with the *M. oryzae* race 007.0. Photographs were taken at 7 dpi. Scale bar, 5 mm. **H** Biomass of *M. oryzae* was measured by qPCR and normalized with rice endogenous *OsUbq*. Relative infection ratio (K59 = 1) is shown. Bars = mean ± s.d. (*n* = 3 independent lines). Different letters indicate significant differences determined by one-way ANOVA (with Tukey's test) (*P* < 0.05). Source data are provided as a Source Data file.

factor(s) determine the functional differences between Pit1 and Pit2.

To identify important residues for Pit1 activity, we generated domain-swapping mutants between Pit1 and Pit2 (Fig. 3A), and transiently expressed them in *N. benthamiana* leaves to observe cell death. The NB-ARC domain appears to determine the functional differences between Pit1 and Pit2 because Pit121 D484V, a Pit1 mutant whose NB-ARC domain is replaced with that of Pit2, failed to induce cell death. Conversely, Pit212 D485V, a Pit2 mutant containing the NB-ARC domain of Pit1, gained cell death activity (Fig. 3A and Supplementary Fig. 8A). We next produced seven more NB-ARC domain-swapping mutants of Pit1, followed by cell death assay in *N. benthamiana* leaves. Pit1-1 (amino acids 290–301 replaced with Pit2), Pit1-4 (amino acids 416–430 replaced with Pit2), and Pit1-6 (amino acids 466–480 replaced with Pit2) dramatically lost their cell death induction activity (Supplementary Fig. 9A). To further narrow down the important amino acids in the NB-ARC domain of Pit1, we generated new Pit1 mutants containing a single Pit2-type amino acid substitution from Pit1-1, Pit1-4, and Pit1-6 and finally revealed that the three residues L301, C416 and G479 are essential for Pit1 D485V-induced cell death (Fig. 3B and Supplementary Fig. 8B). A consistent result showing the importance of L301, C416 and G479 of Pit1 was obtained in cell death assays using rice protoplasts (Supplementary Fig. 9B).

We also examined the effect of mutations at L301, C416, and G479 in Pit1 on disease resistance to rice blast fungus. In this experiment, we used the avirulent rice blast fungus (race 007.0) against Pit1 and rice cultivar Nipponbare, carrying both *Pit1* and *Pit2* genes. The *Pit1* gene is not functional in Nipponbare due to its low expression, making Nipponbare a suitable cultivar to assess exogenous *Pit1* gene function[49]. Plants expressing all three substitution mutants of Pit1 developed a larger lesion against avirulent rice blast fungus compared with that in *Pit1* WT plants, indicating that L301, C416, and G479 are critical residues for Pit1-mediated disease resistance to rice blast fungus (Fig. 3C, D and Supplementary Fig. 9C, D). Notably, the mutations at L301 and C416 led to a more severe defect than that at G479 in cell death induction and disease resistance, indicating that L301 and C416 are more important for Pit1's functions than G479.

The results using the Pit1 mutants demonstrate that L301, C416, and G479 are critical determinants of functional differences between Pit1 and Pit2. We further asked whether introducing these three important residues of Pit1 into Pit2 would make Pit2 function as Pit1. Introduction of one or two Pit1-type residues into Pit2 was not sufficient to induce cell death, but introduction of three Pit1-type amino acids into Pit2 (Pit2 P300L F415C W478G: Pit2 LCG) fully conferred the activity to trigger cell death in *N. benthamiana* leaves (Fig. 3E and Supplementary Fig. 8C) and rice protoplasts (Supplementary Fig. 10A). Next, we tested whether Pit2 LCG complements Pit1 function in disease resistance to two different races of incompatible rice blast fungus. Nipponbare expressing Pit2 LCG inhibited fungal growth to a similar

extent as Pit1 WT, indicating that the Pit2 LCG mutant behaves like Pit1 (Fig. 3F, G and Supplementary Fig. 10B–E).

We performed a Raichu-FRET assay to test the relationship between disease resistance and OsRac1 activation using Pit1 and Pit2 mutants. We found that mutation of all three amino acids in Pit1 D485V abolished the activation of OsRac1 (Fig. 4A and Supplementary Fig. 11A) but Pit2 D484V LCG mutant acquired the ability to activate OsRac1 (Fig. 4B and Supplementary Fig. 11B). Taken together, these results demonstrate that these three residues are critical determinants of the functional differences between Pit1 and Pit2.

## L301 and C416 in Pit1 are important for plasma membrane localization

Because Pit1 and Pit2 displayed different localizations in rice protoplasts (Fig. 2E), we tested whether these important residues contribute to that difference by transient expression assay in *O. sativa* indica *Oc* cells. Pit1 L301P-Venus and Pit1 C416F-Venus lost their plasma membrane localization and were distributed mainly in the cytoplasm, but Pit1 G479W-Venus remained in the membrane (Fig. 4C and Supplementary Fig. 11C). Confirming this altered localization, Pit1 L301P and Pit1 C416F accumulated mainly in the soluble fraction (Fig. 4D). In contrast, the Pit2 LCG mutant acquired substantial but not complete plasma membrane localization (Fig. 4C, D and Supplementary Fig. 11D). About 90% of cells transfected with Pit1 WT-Venus exhibited the plasma localization, whereas 60% of cells expressing Pit2 LCG showed either the plasma membrane or plasma membrane and cytosolic localization (Fig. 4E). These results indicate that L301 and C416 in Pit1 are required for its plasma membrane localization.

## Pit homologs in Poaceae

To understand the evolutionary history of Pit1 and Pit2 proteins, we searched for Pit homologs in the NCBI database using BLASTp and found that they exist only in Poaceae species, including *Sorghum bicolor*, *Setaria italic*, *Panicum miliaceum*, *Cenchrus americanus* and *Triticum aestivum* in addition to *Oryza species*, indicating that *Pit* is a conserved resistance gene in this family (Fig. 5A). Most of the three important residues in Pit homologs correspond to Pit1-type residues, while Pit2-type residues were not observed, implying that an ancestral Pit protein possessed Pit1-type residues at those positions. We further identified *Pit* homologs in 13 domesticated and wild *Oryza* species[34]. AA, BB, and FF are the genome symbols of rice. The genome symbol from A to F is assigned to each rice species based on chromosome pairing during meiosis in the first generation of hybrid germplasm. Phylogenetic analysis revealed that all Pit proteins from 13 *Oryza* species except for *O. brachyantha* (FF genome) belong to Pit1 or Pit2 clades (Supplementary Fig. 12). *O. punctata* (BB genome) has a *Pit2* gene, and all Asian *Oryza* species including *O. nivara*, *O. sativa vg. india*, and *O. sativa vg. japonica* (AA genome) except for *O. rufipogon*, have both *Pit1* and *Pit2* genes (Fig. 5B). Moreover, phylogenetic trees

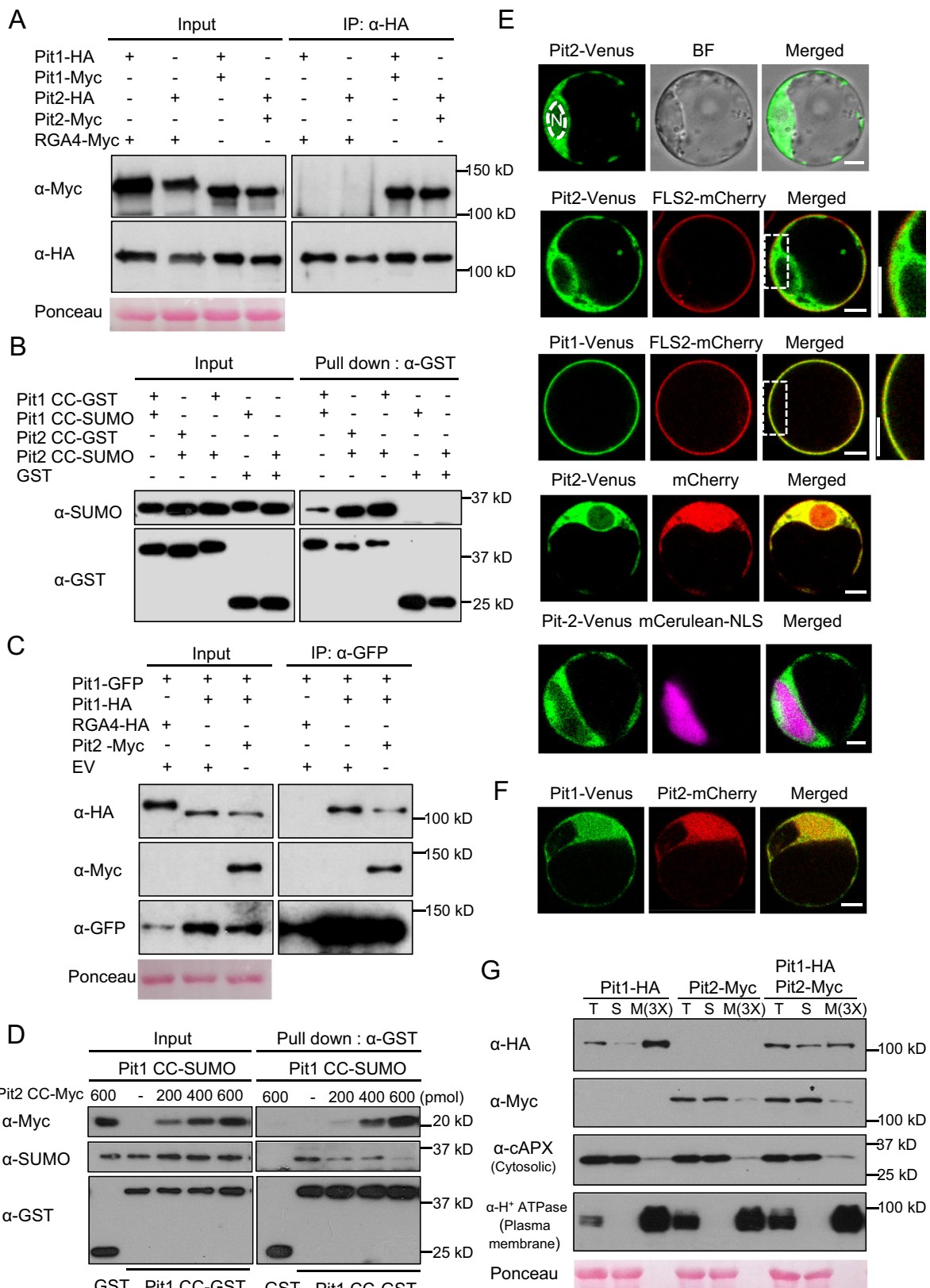

based on the coding sequences of *Pit1* and *Pit2* of *Oryza* species imply that the *Pit* gene was tandemly duplicated before AA genome species expanded (Fig. 5B and Supplementary Figs. 12 and 13). In *O. longistaminata*, gene conversion between Pit1 and Pit2 was observed (Supplementary Fig. 12). Pit2 (KN542601.1) of *O. longistaminata* clustered into Pit1 clade, which is likely due to the gene conversion between Pit1 and Pit2. We employed the formula $T = Ks/2\lambda$ to calculate the duplication and divergence time. $\lambda$ referred to the mutation rate and was considered as $6.5 \times 10^{-9}$ synonymous substitutions per site per year[56]. The two *Pit* genes originated from a duplication within the *Oryza* lineage ~14.69 Mya, before the AA–BB genome divergence of *Oryza* species (Fig. 5A). These observations suggest that the *Pit* gene

**Fig. 2 | Pit2 competes with Pit1 for binding to Pit1 to form a stable hetero-complex mainly in the cytosol. A** In vivo self-association of full-length Pit1 and Pit2. The indicated combinations of tagged proteins were transiently expressed in *N. benthamiana*. Co-IP was performed using anti-HA magnetic beads, and the proteins were detected by Western blot with relevant antibodies. Ponceau staining of Rubisco serves as a loading control. **B** In vitro pull-down assay of homo and heterocomplexes of Pit1 CC and Pit2 CC. GST, Pit1 CC-GST, or Pit2 CC-GST immo-bilized on GST beads was incubated with Pit1 CC-SUMO or Pit2 CC-SUMO. Input and pull-down proteins were detected by Western blot with anti-GST and anti-SUMO antibodies. **C** Pit2 competes with Pit1 to form heteromers in vivo. Pit1-GFP was transiently co-expressed with Pit1-HA in the presence or absence of Pit2-Myc in *N. benthamiana*. Co-IP was performed using anti-GFP agarose beads, and the proteins were detected by Western blot with relevant antibodies. Ponceau staining of Rubisco served as a loading control. **D** Pit2 CC competes with Pit1 CC to form a

heteromer in vitro. Pit1 CC-GST and Pit1 CC-SUMO were co-incubated with anti-GST agarose beads and different amounts of Pit2 CC-Myc. Pull-down assay was carried out using anti-GST beads and proteins were detected by Western blot with corre-sponding antibodies. **E** Localization of Pit1 and Pit2 in rice protoplasts. Protoplasts were transfected with the fluorescent constructs Pit2-Venus with combinations of FLS2-mCherry, mCherry, or mCerulean-NLS. N, nucleus. BF, Bright-field. Scale bars, 5 μm. **F** Colocalization of Pit1 and Pit2 in rice protoplasts. Pit1-Venus was co-transfected with Pit2-mCherry. Scale bars, 5 μm. **G** Cell fractionation assay of Pit1 and Pit2. Pit1-HA, Pit2-Myc, or both were transiently expressed in *N. benthamiana*. Western blot was performed with indicated antibodies. T, S, and M indicate total extract, soluble fraction, and microsomal fraction, respectively. M(3X) indicates three-fold enrichment relative to T and S. Ponceau staining of Rubisco serves as a loading control. All images are representative of results repeated three times with similar results. Source data are provided as a Source Data file.

was tandemly duplicated before the AA–BB genome divergence of *Oryza* species (Fig. 5B), and most of the Asian *Oryza* species kept two *Pit* genes during subsequent evolution. The two residues in Pit1 that are involved in plasma membrane localization are conserved in all Pit1 homologs, showing the importance of plasma membrane localization for Pit1 function. Moreover, Asian *Oryza* Pit2 proteins have at least one substitution that may abolish plasma membrane localization (Figs. 4C and 5B). To examine Pit's function in another genus, we selected *Leersia perrieri* Pit (Pit[per]), which is closest to the *Oryza* Pits (Supple-mentary Fig. 12); it belongs outside the Pit1 and Pit2 clades but has Pit1-type residues at the three important positions. Like Pit1, Pit[per] induced cell death and showed plasma membrane localization in rice proto-plasts (Fig. 5C, D), implying that rice cultivar K59 Pit1 retains the helper function and plasma membrane localization of ancestral Pit.

### Molecular evolution of *Pit1* and *Pit2*
We further analyzed the coding sequences of rice cultivars Nipponbare and K59 as well as the 66-accession pan-genome, which represents all of the major genetically distinct clusters in *O. sativa* and *O. rufipogon*[34,57] (Supplementary Table 2). All 55 of the *O. sativa* cultivars have both genes, while three of 13 *O. rufipogon* accessions have only a *Pit2* gene. Eight cultivars with partial coding sequences or only a *Pit2* gene were excluded from further evolutionary analyses (Supplemen-tary Table 2). The phylogenetic tree of *Pit* alleles divides into two dis-tinct clusters that correspond to the *Pit1* alleles (*Pit1* clade) and *Pit2* alleles (*Pit2* clade), verifying that the duplication predated the diver-gence of *O. rufipogon* and *O. sativa* (Supplementary Figs. 12 and 13).

Genetic diversity is essential for plants to respond to diverse challenges by pathogens[58]. Yang et al. previously analyzed the average nucleotide diversity (π) of 44 NLR genes among 21 rice cultivars and 14 wild rice populations. They categorized the genes into four distinct groups: (1) conserved (π < 0.005), (2) diversified (π > 0.05), (3) intermediate-diversified (π = 0.005−0.05), and (4) present/absent patterns[59]. The average π value for *Pit1* alleles in *O. sativa* was 0.01269, while for *Pit2* alleles, it was 0.01048. This result suggests that the *Pit1* and *Pit2* genes exhibit intermediate levels of nucleotide variation (Supplementary Table 3). Among *O. sativa* species, the value of π was similar to Watterson's nucleotide diversity estimator (θ) in *Pit1*, but π was much higher than θ in *Pit2*, implying that *Pit1* and *Pit2* are under different selection pressures in *O. sativa*. To examine the evolutionary dynamics of *Pit1* and *Pit2* alleles in the 60 *Oryza* accessions, Tajima's *D*, and Fu and Li's test *D**, and *F**, were calculated. Positive values of Taji-ma's *D*, *D**, and *F** in *O. sativa* were obtained for *Pit1*, and *Pit2* had significant positive values of Tajima's *D* and *F**, indicating a clear sig-nature of positive selection on *Pit2*, while *Pit1* does not deviate from neutral evolution. A similar analysis was previously conducted on a pair of NLRs, Pias-1/RGA4, and Pias-2/RGA5[60]. Notably, the sensor Pias-2/RGA5 exhibited a significant positive value for Tajima's *D*. This indi-cates a clear signature of balancing selection imposed on Pias-2/RGA5.

The selection pressure on sensors and helpers appears to vary among each NLR pair. Because the three residues are important for Pit1 and Pit2 function, the *Ka/Ks* values for these three residues were calculated in the *Pit1* and *Pit2* alleles from 53 *O. sativa* cultivars (Table 1, Supple-mentary Table 5, and Supplementary Table 6). The value of *Ka/Ks* exceeded 1 in both P300 and F415, suggesting positive selection for these two residues. Furthermore, the LRR region of Pit2 shows a sig-nificant signature of positive selection in the cultivars *O. sativa* (Sup-plementary Fig. 14 and Supplementary Table 3).

## Discussion
Here, we revealed that Pit2 physically forms stable heteromers with Pit1. Pit2 was the most abundant NLR protein in precipitates with Pit1 D485V-FLAG (Supplementary Table 1). Recent studies have shown that genomically adjacent paired NLRs collaborate to execute immune responses upon pathogen recognition; these comprise rice RGA4-RGA5[28], Pik-1-Pik-2[61], PigmR-PigmS[30] and Pi5-1-Pi5-2[62], and *Arabidopsis* RPS4-RRS1[29], RPP2A-RPP2B[63], SNC1-SIKICs[64] and CHS3-CSA1[37,65–67]. Here, we found that although *Pit1* and *Pit2* are genomically adjacent and share high sequence identity, they have opposite functions in cell death (Fig. 1D, E). This opposite relationship between two NLRs has been reported in several sensor–helper pairs of R proteins, including rice RGA4-RGA5 and PigmR-PigmS as well as *Arabidopsis* RPS4-RRS1[28–30]. How does Pit2 suppress Pit1-induced cell death? We have previously demonstrated that Pit1 is localized in the plasma mem-brane, which plays a key role in Pit1-mediated immunity[50,68]. In *Arabi-dopsis*, the NLR protein ZAR1 forms a pentameric complex named resistosome, which serves as a calcium-permeable channel at the plasma membrane to trigger plant immune signaling[13,14]. The N-terminal sequence of the Pit1 CC domain exhibits similarity to that of the ZAR1 CC domain (Supplementary Fig. 16C). The N-terminal α1 helix of ZAR1 undergoes a conformational switch during resistosome acti-vation, playing a crucial role in Ca[2+] channel function. This finding suggests the potential for Pit1 to share activation mechanisms and functions with ZAR1. Intriguingly, Pit2 interferes with the plasma membrane localization of Pit1, sequestering Pit1 in the cytosol (Fig. 2F, G). This interference by Pit2 likely constitutes the primary mechanism behind its antagonistic effect on Pit1. Moreover, Pit2 can bind to OsSPK1 and OsRac1, which are key downstream molecules for Pit1, at similar levels to Pit1[50,52] (Supplementary Fig. 7), but Pit2 is unable to trigger OsRac1 activation and immune responses (Fig. 1C). Pit2 com-petitively attenuates Pit1 homocomplex formation by titrating out Pit1 to form a heterocomplex with Pit1 (Fig. 2C, D). Pit2 may compete with Pit1 for binding to OsSPK1 and OsRac1, acting in a dominant negative manner.

Although the transient knockdown of *Pit2* triggered Pit1-mediated cell death in rice protoplasts (Fig. 1F), we could not observe the clear suppressor function of Pit2 toward Pit1 in *Pit2* KO of rice cultivar K59 (Fig. 1G and H and Supplementary Fig. 5C). Okuyama et al. also

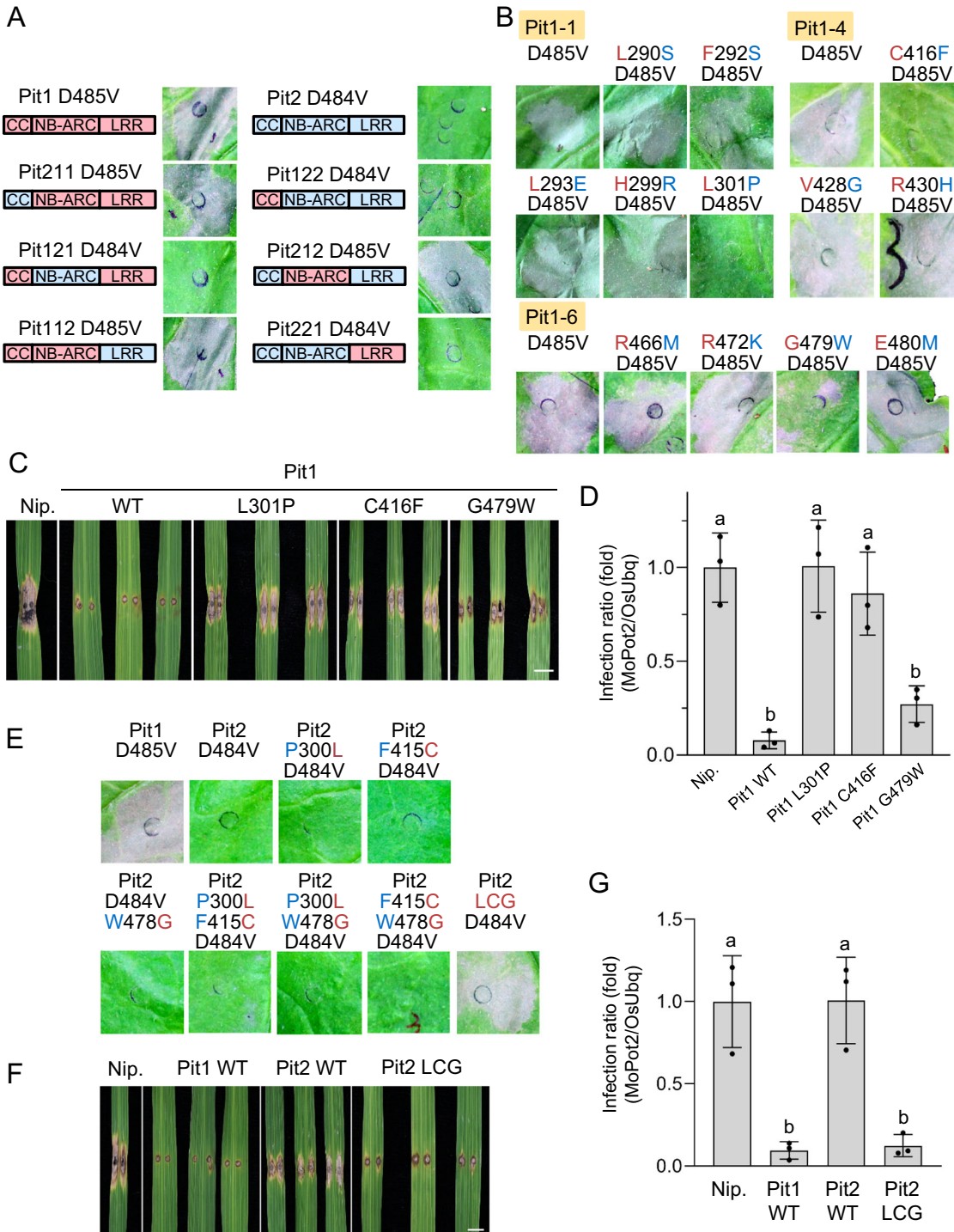

**Fig. 3 | Three residues in the NB-ARC domain determine the functional difference between Pit1 and Pit2. A** Schematic architecture and cell death phenotype of Pit1 D485V, Pit2 D484V and domain-swapping mutants. Red and blue indicate domains derived from Pit1 and Pit2, respectively. The indicated proteins were transiently expressed in *N. benthamiana*, and cell death was photographed at 3 dpi. **B** Cell death phenotype of Pit1 D485V and the indicated amino acid substitution mutants in *N. benthamiana*, photographed at 3 dpi. Red and blue characters indicate residues derived from Pit1 and Pit2, respectively. **C** Infection assays of Pit1 mutant plants with the incompatible *M. oryzae* race 007.0. Photographs show phenotypes of representative Pit1 WT and Pit1 mutant plants at 7 dpi. Scale bar, 5 mm. **D** Biomass of the incompatible *M. oryzae* race was measured by qPCR and normalized with endogenous *OsUbq*. Relative infection ratio (Nipponbare (Nip.) = 1) is shown. Bars represent the mean ± s.d. (*n* = 3 independent lines). Different letters above bars indicate a significant difference determined by one-way ANOVA (with Tukey's test) (*P* < 0.05). **E** Cell death phenotype of Pit2 D484V and Pit2 mutants in *N. benthamiana*, photographed at 3 dpi. Red and blue characters indicate residues derived from Pit1 and Pit2, respectively. **F** Infection assays of Nipponbare expressing *Pit2* mutants with the incompatible *M. oryzae* race 007.0. Photographs show phenotypes of representative Pit1 WT, Pit2 WT, and Pit2 LCG plants at 7 dpi. Scale bar, 5 mm. **G** Biomass of the incompatible *M. oryzae* race was measured by qPCR and normalized with endogenous *OsUbq*. Relative infection ratio (Nip. = 1) is shown. Bars represent the mean ± s.d. (*n* = 3 independent lines). Different letters above bars indicate a significant difference determined by one-way ANOVA (with Tukey's test) (*P* < 0.01). Source data are provided as a Source Data file.

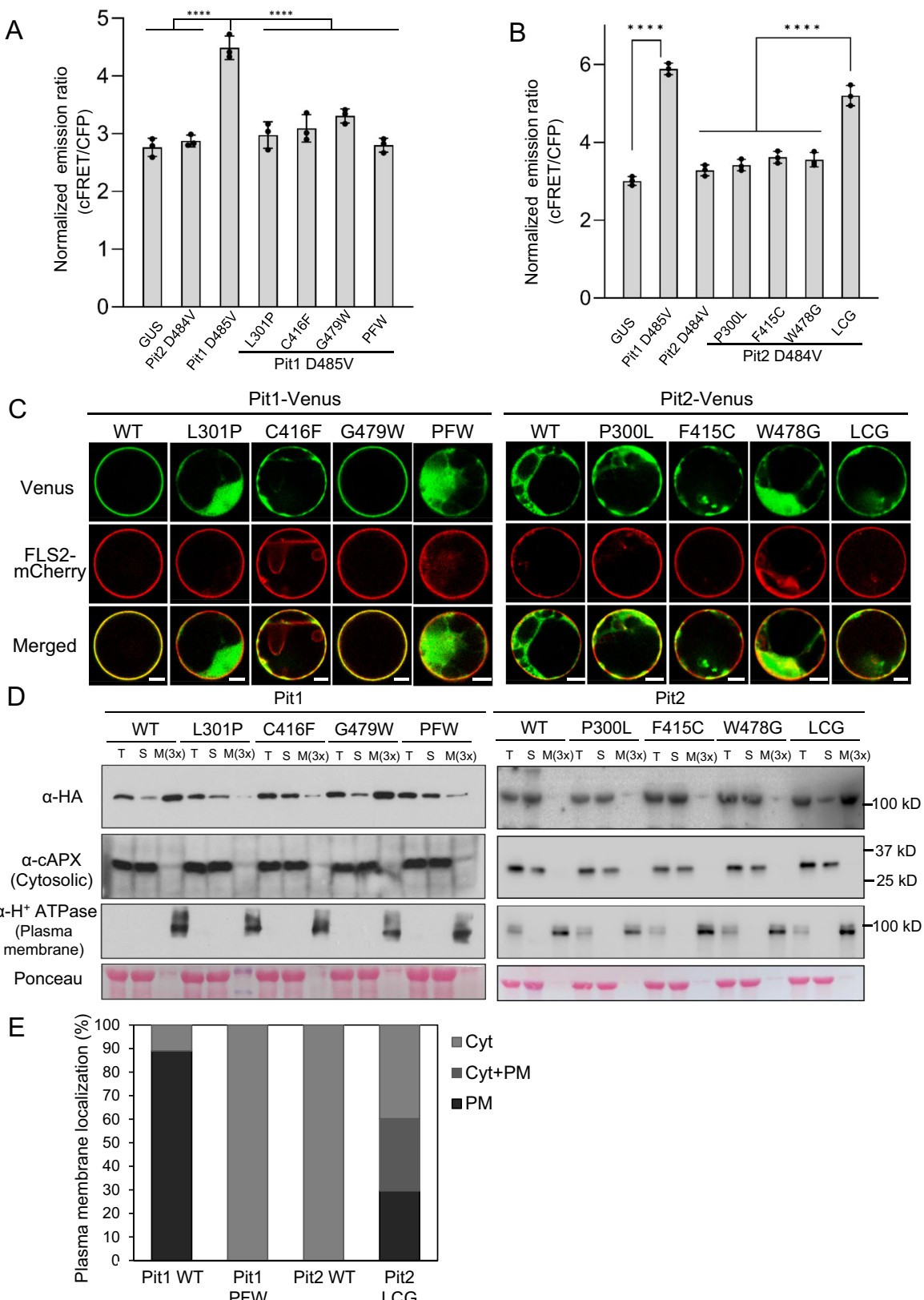

reported a similar result that the introduction of only the helper NLR *RGA4* in a rice cultivar Kanto 51, which lacks endogenous *RGA4* and *RGA5* genes, does not induce visible autoimmunity[69]. The phenotype in *Pit2* KO of rice cultivar K59 might be due to compensation from PAN1–PAN5 because we identified five NLRs PAN1–PAN5 that also co-immunoprecipitated with Pit1 (Supplementary Table 1). We revealed

that PAN3 also contributes to Pit1-mediated cell death (Supplementary Fig. 4G) and the transient knockdown of *Pit2* enhanced the expression of *PAN3* in K59 rice suspension cells (Supplementary Fig. 4F), implying that *Pit2* is genetically linked to *PAN3* and *Pit2* KO possibly leads to the increment of *PAN3* expression, resulting in the compensation of *Pit2* KO phenotype.

**Fig. 4 | Pit1 L301 and C416 are important for OsRac1 activation and plasma membrane localization. A**, **B** Monitoring OsRac1 activation by amino acid-substituted Pit1 (**A**) and Pit2 (**B**) mutants using Raichu-OsRac1 FRET in vivo. Statistical analysis of OsRac1 activation by Raichu-OsRac1 with normalized emission ratios of Venus to CFP. PFW, Pit1 L301P C416F G479W. Bars represent the mean ± s.d. (*n* = 3 biological replicates). The asterisks indicate significant differences determined by one-way ANOVA (with Tukey's test) (***$P < 0.0001$). **C** Localization of Pit1, Pit2, and the indicated amino acid substitution mutants in rice protoplasts. Protoplasts were co-transfected with the indicated Venus-tagged fluorescent constructs and the plasma membrane marker FLS2-mCherry. Scale bars, 5 μm. This experiment was repeated three times with similar results. **D** Cell fractionation assay

showing the localization of Pit1, Pit2, and the indicated mutants; the HA-tagged proteins were transiently expressed in *N. benthamiana*. Western blot was performed with anti-HA, anti-cAPX (cytosolic marker), and anti-H⁺ATPase (plasma membrane marker) antibodies. M(3x) is three times enrichment relative to T or S. T, S, and M indicate total extract, soluble fraction, and microsomal fraction, respectively. Ponceau staining of Rubisco served as a loading control. This experiment was repeated three times with similar results. **E** Quantification of localization of Pit1, Pit1 PFW, Pit2 and Pit2 LCG. One hundred transfected cells were counted under a microscope. PM, plasma membrane. Cyt, Cytosol. Source data are provided as a Source Data file.

Domain-swapping experiments revealed that L301 and C416 in the NB-ARC domain of Pit1 are essential for plasma membrane localization and disease resistance (Fig. 3). Characteristic residues similar to L301 and C416 occur in other NLR proteins (Supplementary Fig. 15). L301 and C416 are located in the resistance nucleotide-binding site (RNBS)-B and RNBS-D motifs, respectively, which are highly conserved in NLR family proteins (Supplementary Fig. 3A)[6]. Mutations in the RNBS-B and RNBS-D motifs of various NLR proteins abolish cell death and disease resistance, demonstrating the importance of these motifs in the activation of NLRs (Figs. 3 and 4)[6,70]. Pit2 LCG which contains three Pit1-type amino acids fully complements Pit1 function in resistance to rice blast fungus (Fig. 3F, G), and L301 and C416 in Pit1 are more important for Pit1 functions than G479 (Supplementary Fig. 9B, D). The mutations at the position of P300 and F415 in Pit2 are probably the fate-determining mutations between Pit1 and Pit2 (Fig. 6). Pit2 has produced a dominant negative effect on Pit1-induced cell death, and Pit2 is thus the paralogue of Pit1 (Fig. 1). This phenomenon is called paralogue interference, in which a physical and functional link in the paralogous heteromers produces a dominant negative effect by mutations, thereby contributing to shaping the fate of the duplicates[21,71].

To understand the function of L301 and C416 in Pit1, we generated a model structure of Pit1 based on the active (dATP-bound) and inactive (ADP-bound) NLR ZAR1 structure[11,12,14] (Supplementary Fig. 16A, B). Our model predicts that the side chains of L301 and C416 residues are buried in the protein interior. L301 in Pit1 appears to contribute to the stability of ATP-bound form, which is attributable to hydrogen bonds between dATP and the NB domain. In the active ZAR1 structure, R297, which is located in a short loop between β sheet 4 (β4) and 3¹⁰ helices, forms a hydrogen bond with the γ-phosphate group of dATP. In the model structure of active Pit1 R305 (corresponding to R297 in ZAR1) has a hydrogen bond with the γ-phosphate group of dATP (Supplementary Fig. 16B). L301 is in β4, which connects to the short loop containing R305 and the side chain of L301, and has a hydrophobic contact with α-helix 6 (α6), which is involved in hydrogen bonds with the α- and β-phosphate group of dATP. Thus, dATP seems to be stabilized by the combination of the hydrogen bond of R305 to the γ-phosphate group of dATP and those of α6 to the α- and β-phosphate group of dATP. The hydrophobic contact between β4 containing L301 and α6 may contribute to this ATP-bound form stability. The significance of L301 in ATP-bound form stability is further supported by this hydrophobic contact and the L-to-P mutation at L301 of Pit1 may result in loss of the hydrophobic contact between β4 and α6, leading to a decrease in the ATP-bound form stability. Supporting this speculation, we have previously found that a P-loop mutant of Pit1, which lacks ATP binding activity, does not localize to the plasma membrane[50]. Similarly, P-loop mutants of several plasma membrane-localized NLR proteins such as RPM1 and TM-2² lose their ability to tether to the plasma membrane and to trigger immune induction[47,72]. Pit1 C416 may contribute to CC domain dynamics. ZAR1 appears to be kept in an inactive state via contacts among LRR, helical domain 1 (HD1), and winged-helix domain (WHD)[12,14]. The activation of ZAR1 induces the formation of a wheel-like pentamer complex, triggered by

a structural change in α1 of the ZAR1 CC domain. The primary sequence of α1 of the Pit1 CC domain shows similarity to that of the ZAR1 CC domain (Supplementary Fig. 16C). C416 in Pit1 is located in α14 of the WHD domain (Supplementary Fig. 16A, B). We note that α14 appears to form a hydrophobic contact with two α-helices, α15 and α16, which stabilizes WHD conformation. This tight conformation of the WHD domain packs together LRR, α1 of the CC domain, and the WHD domain itself, and the packing among CC-WHD-LRR renders Pit1 inactive. The C-to-F mutation in Pit1 may increase the molecular packing density of the three α-helices (α14, α15, and α16) in the WHD domain, leading to a reduction of structural flexibility of the CC-WHD-LRR conformation that keeps Pit1 in an inactive form.

We assume a possible evolutionary model of *Pit1* and *Pit2* as follows: the duplication of an ancestral *Pit* gene occurred about 14.69 Mya. Judging from the three important residues (LCG), *Pit2* is considered a new copy (Figs. 5A and 6). Either *Pit1* or *Pit2* gene may be enough to fulfill resistance to rice blast fungus or other pathogens, and therefore the other gene was redundant. Therefore, in *O. punctata, O. meridionalis, O. glaberrima* and *O. barthii*, one of them could have been deleted (Fig. 5B). The heteromerization between Pit1 and Pit2 exhibited significantly greater strength compared to the homomeric association of Pit1 in the in vitro binding assay (Fig. 2B), and Pit2 emerged as the predominant NLR within the array of NLRs that precipitate alongside Pit1 (Table 1). This observation implies the formation of a robust and stable heteromer complex between Pit1 and Pit2 in *planta*. Occasionally, as the redundant, Pit2 accumulated substitutions including the fate-determining mutation between Pit1 and Pit2. The mutations at P300 and F415 in Pit2 have resulted in neo-functionalization, causing the loss of its plasma membrane localization and acquisition of a new function. This new function sequesters Pit1 in the cytosol and regulates the membrane-localized Pit1 functions such as cell death execution (Fig. 6). To support this hypothesis, the Ka/Ks analysis on the three important residues for the functional difference between Pit1 and Pit2 and the selection server program on Pit2 revealed that P300 and F415 in Pit2 are under positive selection (Supplementary Table 3 and Supplementary Fig. 14). What is the significance of Pit2 forming a complex with Pit1? Interestingly, all the accessions of domesticated rice *O. sativa Indica* and *Japonica* possess both *Pit1* and *Pit2* genes, suggesting that the fixation of *Pit1* and *Pit2* occurred during rice domestication in Asia (Fig. 5B and Supplementary Table 2). Notably, a well-established trade-off exists between yield and disease resistance (also called fitness costs of NLR-mediated resistance). Two field experiments with *Arabidopsis* isogenic lines, carrying NLR gene *RPM1* or *RPS5*, have consistently demonstrated a significant reduction in seed production compared to their NLR-eliminated counterparts[73,74]. One interesting hypothesis is that Pit2 acts as a fine tuner of the membrane-localized Pit1 functions, ensuring that immune activation is carefully regulated to prevent excessive response. It is possible that this regulatory mechanism potentially enhances the yield of domesticated rice. If we overexpress Pit2 expression to sufficiently suppress the activity of Pit1, compromising the Pit1-mediated disease resistance and increasing rice yield, we may strengthen our hypothesis. Subsequent investigations into the intricate relationship between rice

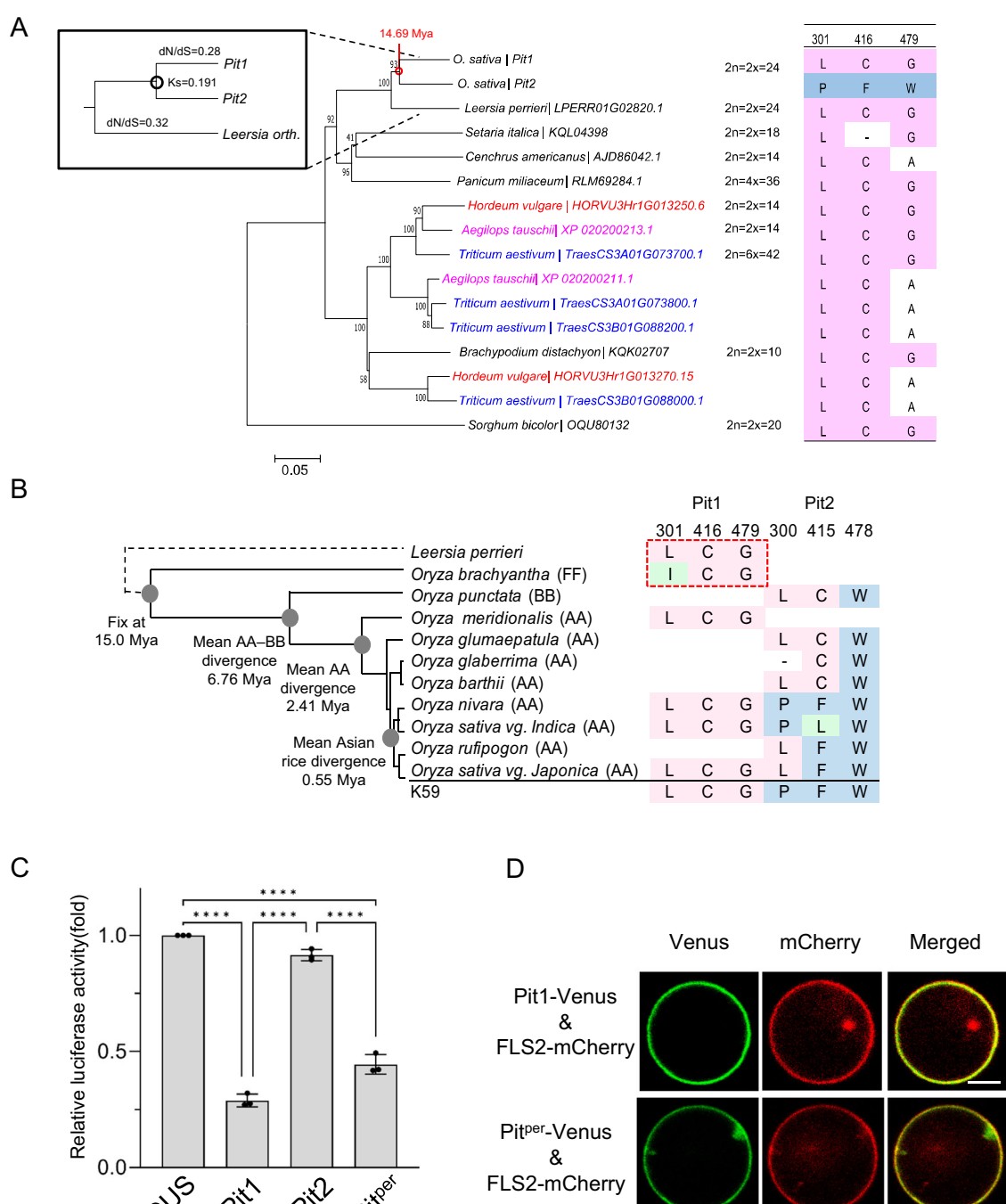

**Fig. 5 | Pit homologs in *O. sativa* and other species. A** Phylogenetic analysis of Pit homologs in Poaceae species. Three residues corresponding to L301, C416 and G479 of Pit1 in different homologs are shown to the right. The inset highlights the estimated duplication time of *Pit1* and *Pit2*. **B** The three residues corresponding to L301, C416 and G479 of Pit1 in 13 domesticated and wild rice species. The red dotted box indicates that *Pit* genes in *L. perrieri* and *O. brachyantha* are outside the Pit1 and Pit2 clades (Supplementary Fig. 12). This phylogenetic tree is adapted from Stein et al.[34]. **C** Cell death activity of Pit from *L. perrieri* in rice protoplasts. The indicated constructs were co-transfected with a luciferase reporter vector in rice protoplasts,

and LUC activity was measured at 40 h after transfection. Relative luciferase activity (GUS = 1) is shown. Bars represent the mean ± s.d. (*n* = 3 biological replicates). The asterisks indicate significant differences determined by one-way ANOVA (with Tukey's test) (\*\*\*\**P* < 0.0001). **D** Localization of Pit from *L. perrieri* in rice protoplasts. Protoplasts were transfected with the fluorescent constructs Pit1-Venus or Pit^per-Venus in combination with the control fluorescent protein FLS2-mCherry, and fluorescence in the protoplasts was observed at 12–16 h after transfection. Scale bars, 5 μm. Source data are provided as a Source Data file.

domestication and the Pit1-Pit2 pair are crucial for a comprehensive understanding.

In addition, we found the selection signatures at the C-terminus of Pit2: (1) The Tajima's *D* value in the LRR domain of *Pit2* in *O. sativa* and *O. sativa japonica* is positive (Supplementary Table 3). (2) The ML tree of Pit2 divides into two major clusters (Supplementary Fig. 13). (3) The

selection server program shows that Pit2 accumulates the positively selected residues at its C-terminal LRR domain (Supplementary Fig. 14). *Pit2* KO partially abrogated resistance to avirulent rice blast fungus (Fig. 1H). These results raise the possibility that the polymorphisms in the LRR domain of Pit2 contribute to the specific recognition of pathogen effectors. An alternative hypothesis is that

**Table 1 | Ka/Ks in values for three amino acids of Pit1 and Pit2 alleles as determined by the M5, M8, and MEC models**

| Group | Model | Pit1 | | | Pit2 | | |
|---|---|---|---|---|---|---|---|
| | | Ka/Ks value for aa site | | | Ka/Ks value for aa site | | |
| | | L301 | C416 | G479 | P300 | F415 | W478 |
| *O. sativa* | M5 | 0.36 | 0.4 | 1.6 | 5.3* | 6.5* | 0.26 |
| | M8 | 0.29 | 0.37 | 2.7 | 4.8* | 4.9* | 0.16 |
| | MEC | 0.34 | 0.38 | 1.4 | 15* | 18* | 1.3 |
| *O. sativa japonica* | M5 | 0.65 | 0.65 | 0.65 | 4.1 | 4.9* | 0.23 |
| | M8 | 0.79 | 0.8 | 0.8 | 4.1 | 5.9* | 0.44 |
| | MEC | 1 | 1 | 1 | 11 | 16* | 1.6 |
| *O. sativa indica* | M5 | 0.33 | 0.37 | 1.9 | 3.4 | 5.6* | 0.32 |
| | M8 | 0.27 | 0.34 | 2.7 | 3.6 | 4.8* | 0.15 |
| | MEC | 0.32 | 0.36 | 1.3 | 8.5 | 14* | 1.3 |

Positive selection sites having statistical significance (the lower boundary of confidence interval > 1) under the Bayesian test were used for calculation. * indicates the lower boundary of confidence interval > 1.

Pit2 fine-tunes Pit1 functions, and PAN1-5 acts as sensors for effectors from *M. oryzae* and other pathogens. Since Avr-Pit has not yet been identified, we would like to address how Pit1 and Pit2 recognize Avr-Pit in future studies.

## Methods

### Plant materials and growth conditions

*Nicotiana benthamiana* plants were grown at 24 °C, 50% relative humidity with a 16/8 h light/dark photoperiod for 3 weeks. Rice cultivar K59 carries functional *Pit1* and *Pit2* genes. *Oryza sativa* japonica Nipponbare and *Oryza sativa* indica Oc possess both *Pit1* and *Pit2* genes, but *Pit1* is non-functional due to low expression. Rice plants were reared in a growth room at 28 °C, 60% relative humidity with a 12/12 h light/dark photoperiod. Rice suspension cells were cultured at 30 °C with shaking at 100 rpm.

### Plasmid construction

Primers used in this study are listed (Supplementary Table 4). The relevant genes or DNA fragments were cloned into the pENTR D-TOPO vector (Invitrogen), and depending on the requirement of the experiments they were transferred into destination vectors by LR reaction. The destination vectors used were pGWBs (for cell death, protein expression, and Co-IP in *N. benthamiana*), Ubq-GW (for cell death, protein expression, and Co-IP in rice protoplasts), pBTM116 and pVP16 (for Y2H), p2k (for overexpression in rice) and p1300-(Cas9) (for expression or knockout in rice), pSUMO and pCold-GST (for protein purification), and p2k-pANDA mini (for transient suppression in rice protoplasts).

### Site-directed mutagenesis

Overlap extension PCR primers containing mutation sites were designed for generating site-directed mutations (Supplementary Table 4). PCR was performed by KOD-Plus-Neo (ToYoBo) using pENTR-Pit1 or 2 as a template, followed by digestion of the products with *DpnI* (NEB) and the digested plasmids were transformed into *Escherichia coli* (DH5α).

### Co-IP and LC-MS/MS Analysis

After 100 µM estradiol treatment, rice suspension cells were harvested and ground in a protein extraction buffer [10 mM Tris·HCl (pH 7.5), 150 mM NaCl, 1 % Nonidet P-40 and 1 mM EDTA (pH 7.5)]. Crude homogenates were centrifuged at 4 °C at 20,000 $g$ for 30 min to remove cellular debris. The supernatants were incubated with anti-FLAG (F3165; Sigma) antibody-immobilized with Dynabeads Protein G

(10004D: VERITAS) for 1 h with rotating. After 3 times washing with the protein extraction buffer, bound proteins were eluted by the addition of 0.25 mg/ml FLAG peptide. The resultant elutions were subjected to SDS-PAGE. LC-MS/MS analysis was performed using an HTC-PAL/Paradigm MS4 system coupled to an LTQ Orbitrap XL (Thermo Scientific) mass spectrometer. Trypsin-digested peptides were loaded on the L-column (100 mm internal diameter, 15 cm; CERI) using a Paradigm MS4 HPLC pump (Michrome BioResources) and an HTC-PAL autosampler (CTC Analytics). Buffers were 0.1% (v/v) acetic acid and 2% (v/v) acetonitrile in water (Solvent A) and 0.1% (v/v) acetic acid and 90% (v/v) acetonitrile in water (Solvent B). A linear gradient from 5 to 45% buffer B, of 26 min duration, was applied, and peptides eluted from the L-column were introduced directly into an LTQ Orbitrap XL mass spectrometer with a flow rate of 500 nL/min and a spray voltage of 2.0 kV. All events for the MS scan were controlled and acquired by Xcalibur software version 2.0.7 (Thermo Scientific). The range of MS scan was m/z 400 to 1500 and the top three peaks were subjected to MS/MS analysis. Obtained spectra were compared with a protein database (NCBI, Taxonomy; *Oryza sativa*) using the MASCOT server (version 2.2). The mascot search parameters were as follows: peptide tolerance at 10 ppm, MS/MS tolerance at ±0.8 Da, peptide charge of 2+ or 3+, trypsin as enzyme allowing up to five missed cleavage, carbamidomethylation on cysteine as a fixed modification, and oxidation on methionine and phosphorylation on serine and threonine as a variable modification.

### Yeast two-hybrid assay

To test protein interactions in yeast, L40 cells were transformed with pBTM116 or pVP16 constructs. Co-transformants were plated on synthetic medium containing or lacking histidine, and incubated at 30 °C for 3 days.

### *Agrobacterium*-mediated transient expression in *N. benthamiana* leaves

Agroinfiltration of *N. benthamiana* leaves was performed as described previously[52]. *Agrobacterium tumefaciens* strain GV3101 was used to infiltrate the leaves of 3-week-old *N. benthamiana* plants. We also used the p19 silencing suppressor to enhance gene expression. Transformed GV3101 cells were grown overnight to an optical density at 600 nm (OD$_{600}$) of -0.8. The bacteria were collected by centrifugation and resuspended in buffer [10 mM MgCl$_2$, 10 mM MES-NaOH (pH 5.6), and 150 µM acetosyringone], adjusted to OD$_{600}$ = 0.4, and incubated at room temperature for 2 h before infiltration. After infiltration, plants were reared in a growth room for protein expression and cell death assays.

### Cell death assay in *N. benthamiana*

Transformed *A. tumefaciens* strains expressing Pit1/2 WT or the mutated genes were infiltrated into *N. benthamiana* leaves by the method described above. Each strain was infiltrated in a 1 cm diameter circle on 15 leaves for three independent experiments. At 2–3 dpi, cell death phenotype was observed.

### Protein expression and co-IP assay

Protein extraction from *N. benthamiana* and rice leaves and Co-IP assay were performed as described previously[52]. For the Co-IP assay with rice protoplasts, a large-scale protoplast transformation was used. Each plasmid (50 µg) was added to 1 mL (5 × 10⁶ cells/mL) rice protoplasts and incubated at 30 °C for 16 h. Proteins were extracted using IP buffer [20 mM Tris·HCl (pH 7.5), 150 mM NaCl, 10% glycerol, 1 mM EGTA (pH 7.5), 5 mM DTT, 0.2% Nonidet P-40, and EDTA-free protease inhibitor (Roche)] at 4 °C for 2 h and then centrifuged at 21,500 × $g$ for 15 min at 4 °C. A 40 µL aliquot of each supernatant was used as input for Western blot analysis and the remainder was incubated with 15 µL anti-FLAG agarose beads (SLBW2732; Sigma) for 1 h at 4 °C. The

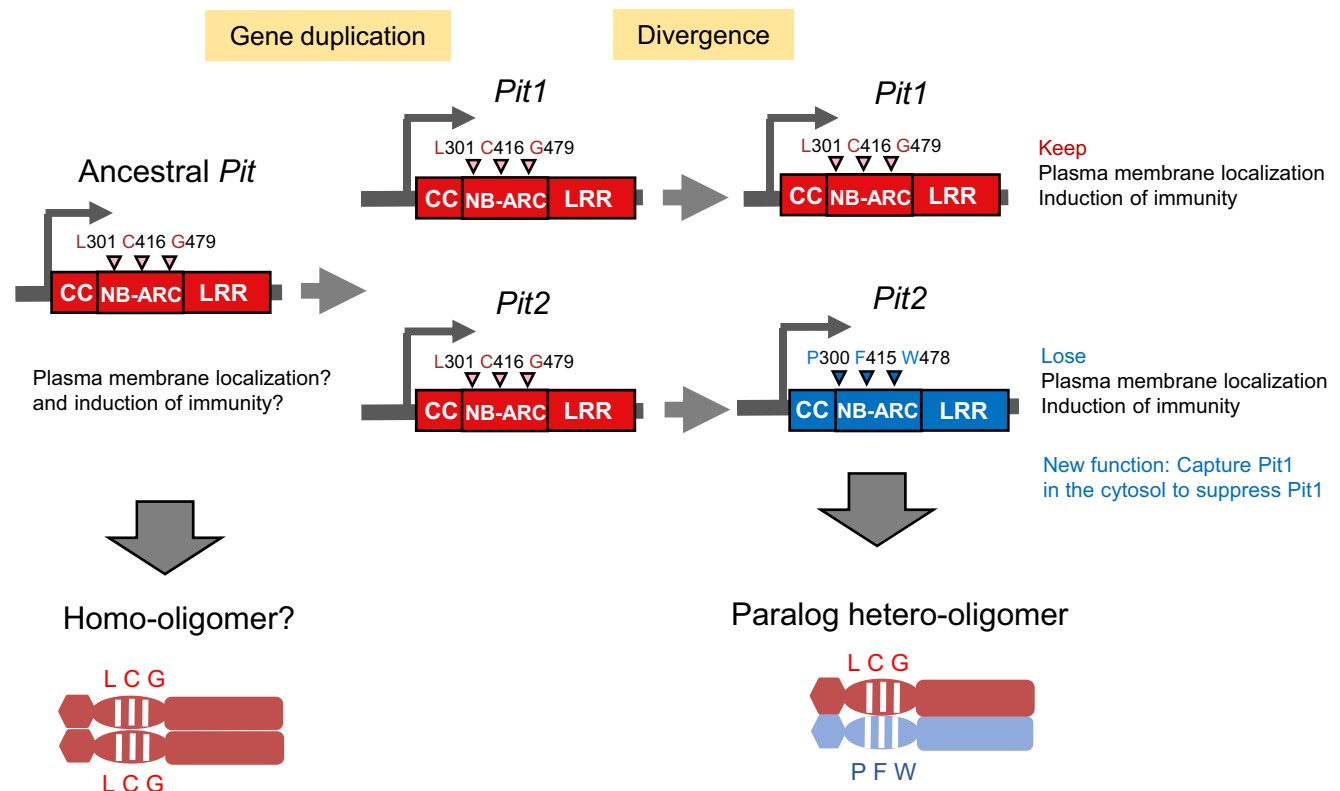

**Fig. 6 | Model of the evolution and function of Pit1 and Pit2.** The ancient *Pit* gene is a conserved resistance gene in Poaceae species and appears to have been duplicated in *Oryza* species. After duplication, *Pit2* is under positive selection and the three important residues have evolved from LCG in Pit1 to PFW in Pit2, leading to Pit2 losing the ability to localize in the plasma membrane but gaining a new function to suppress Pit1-mediated cell death.

samples were washed 5 times for 5 min each with IP buffer. Immune complexes were eluted in 100 μL of 1× SDS loading buffer by heating at 95 °C for 10 min. The IP and input samples were subjected to immunoblotting with anti-c-Myc (9B11; Cell Signaling Technology), anti-cAPX (AS06 180; Agrisera), anti-H⁺ATPase (AS07 260; Agrisera) and anti-FLAG (F3165; Sigma) antibodies.

## Cell fractionation

Microsomal fractionation from *N. benthamiana* leaves was based on methods reported previously[75]. Briefly, *N. benthamiana* leaves (about 300 mg) expressing proteins were frozen in liquid nitrogen, and ground to fine powder with a pestle and mortar. One milliliter of ice-cold sucrose buffer [20 mM Tris·HCl (pH 8.0), 0.33 M sucrose, 1 mM EDTA (pH 8.0), 5 mM DTT and EDTA-free protease inhibitor (Roche)] was added. Samples were mixed and centrifuged at $2,000 \times g$ for 5 min at 4 °C to remove cell debris. A 100 μL portion of the supernatant was used as the total lysate (T), and the remainder was transferred to a new tube and centrifuged at $20,000 \times g$ for 60 min at 4 °C; 100 μL supernatant was then used as the soluble fraction (S). The pellet was resuspended in 300 μL sucrose buffer as the microsomal fraction (M).

## RNA extraction and qRT-PCR

Total RNA was extracted with TRIzol reagent (Invitrogen) according to the manufacturer's instructions. RNA was reverse-transcribed with a cDNA synthesis kit (Vazyme) according to the manufacturer's protocol. qRT-PCR was performed using SYBR Green Supermix (Bio-Rad) in a CFX96 Touch Real-Time PCR Detection System (Bio-Rad). *OsUbiquitin* served as an internal control. Primer sequences for qRT-PCR and RT-PCR are shown in Supplementary Table 4.

## Luciferase activity assay in rice protoplasts

Protoplasts were prepared from rice suspension cells as described previously[55]. To monitor protoplast viability, the firefly *luciferase* (*LUC*) reporter gene was expressed under the maize *Ubiquitin* promoter. *Pit WT* or *Pit* mutant plasmids were co-transfected with 2 μg of *LUC* plasmid into rice protoplasts ($5 \times 10^6$ cells/mL) by the polyethylene glycol method. Transformed protoplasts were incubated for 40 h at 30 °C, and proteins were extracted in 1× lysis buffer provided in the Luciferase Assay Report Kit (Promega) followed by centrifugation at $21,500 \times g$ at 4 °C for 1 min. The supernatant (20 μL) was then mixed with 100 μL luciferase substrate and luciferase activity was measured in a microplate reader. Luminescence values were normalized using protoplasts expressing GUS. The experiments were repeated three times independently.

## Subcellular localization

Protoplast isolation and transformation were as described above. After transfection and incubation for 12–18 h at 30 °C, the protoplasts were examined under a Leica TCS-SP8 microscope. Fluorescence signals from Venus and mCherry were captured by sequential excitations with 514- and 598-nm lasers, respectively.

## Protein purification and in vitro pull-down assays

GST/His-fused proteins were expressed in *E. coli* strain BL21(DE3). Transformed cells were grown in Luria-Bertani liquid medium at 37 °C to $OD_{600}$ ~ 0.6, and protein expressions were induced with a final concentration of 0.3 mM isopropyl-β-D-thiogalactopyranoside for 12–16 h at 16 °C. The bacteria were collected by centrifugation and sonicated in homogenization buffer [20 mM Tris·HCl (pH 8.0), 150 mM NaCl, 1 mM DTT]. After centrifugation at $21,500 \times g$ for 1 h, the

supernatants were purified by affinity chromatography using Ni-NTA agarose resin (Clontech) or Glutathione Sepharose 4B (GE Healthcare).

For in vitro binding assay, equal amounts (1 nmol) of His-SUMO and GST fusion proteins were mixed with Glutathione Sepharose 4B beads in 200 μL pull-down buffer [50 mM Tris·HCl (pH 7.5), 150 mM NaCl, 0.1% Triton X-100, 1 mM EDTA and 5 mM DTT], and incubated with rotation at 4 °C for 30 min. The beads were then washed 5 times with a pull-down buffer. For the in vitro competitive binding of Pit2 with Pit1 homooligomer, Pit1-CC-GST, Pit1-CC-SUMO, and Pit2-CC-Myc were prepared at the same concentration of 10 μM. Briefly, Pit1-CC-GST and Pit1-CC-SUMO (600 pmol) were co-incubated with anti-GST beads for 20 min. Different amounts of Pit2-CC-Myc (0, 200, 400, and 600 pmol) were then added and the mixture was incubated for 20 min. Bound proteins were eluted with 100 μL 1 × SDS loading buffer and subjected to immunoblot assay with anti-GST (sc-138; Santa Cruz Biotechnology), anti-SUMO (A01693; GenScript) and anti-Myc antibodies.

### Raichu-OsRac1 FRET analysis
To monitor the activation of OsRac1 by Pit in vivo, we used the Raichu intramolecular FRET system, as reported previously[55]. Rice protoplasts were isolated and transfected as described above. Transformed protoplasts were imaged using a Leica SMD FLCS confocal microscope. Raichu-OsRac1 was excited using a 440-nm solid-state laser. The CFP and Venus emission filters were set at $470 \pm 20$ nm and $550 \pm 25$ nm, respectively.

### Targeted mutagenesis of *Pit1* and *Pit2* in rice with CRISPR/Cas9
The *Pit1* and *Pit2* genes in the K59 cultivar were targeted with two gRNA spacers in the N-terminus (within the first 450 bp) of each gene (Supplementary Table 4). The gRNA genes were fused into a binary vector (pKO-Pit1/2) by T4 Ligase (New England Biolabs, USA) and followed with the transformation events in rice. Mutation sites were detected by PCR with specific primers in the flank of the two target sites of each gene.

### Rice infection with rice blast fungus
An infection assay using *M. oryzae* strain Ina86-137 (Race 007.0) or others was performed on japonica rice cultivars K59 and Nipponbare[49]. Blast fungus growth and punch infection of leaf blades were described previously[50]. In brief, blast fungus was grown on oatmeal agar plates at 23 °C for 14 days in the dark. After washing the fungus using sterile water, the plates were incubated under white light for another 14 days. For punch infection, the two or three youngest fully developed leaves from two-month-old plants were punched at 1.5 mm diameter and a piece of oatmeal agar containing blast fungus was attached to the hole. Lesion length measurements and photographs of disease lesions were taken at seven days after inoculation.

### Evolutionary analysis of Pit
Pit1 and Pit2 were used as query sequences for BLASTp searches against the NCBI database and 13 *Oryza* species' genomes to find homologs with e-value 1e-5[34]. The orthologous genes were subjected to in-codon frame alignment sequences using MAFFT. Based on alignment sequences, phylogenetic trees were constructed by MEGA V6.0 software using neighbor-joining (NJ) with a p-distance model. Missing data and gaps were processed by pairwise deletion, with the bootstrap value set to 1000. To estimate the divergence time between Pit1 and Pit2, the synonymous substitution ratio (Ks) was calculated using DnaSP V6.0, with an estimated rate of $6.5 \times 10^9$ substitutions per synonymous site per year[76]. Pit1 and Pit2 were used as query sequences for BLASTp searches against the rice Pan-genome accession[57]. The CDS sequences were aligned by using MAFFT. Average nucleotide polymorphism ($\pi$) and Watterson's estimator ($\theta$) of *Pit1* alleles and *Pit2* alleles in 60 cultivars were calculated using DnaSP V6.0[57,77].

Phylogenetic analysis of *Pit1* and *Pit2* in 60 rice accessions was performed with the maximum likelihood method (ML) method in PhyML-3.1. The possibility of selection on the *Pit* gene was examined with Tajima's $D$ and Fu and Li's $D*$ and $F*$ using DnaSP V6.0[77], and the Selection Server program (http://selection.tau.ac.il) was then used to determine the value of Ka/Ks for three amino acids of interest in the *Pit2* alleles from 53 *O. sativa* cultivars.

### Statistical analysis
All data in this study were analyzed using GraphPad Prism 9. One-way (with Tukey's test) or two-way (with Tukey's test or Šídák's test) analysis of variance (ANOVA) for multiple comparisons was used to determine statistical significance.

### Reporting summary
Further information on research design is available in the Nature Portfolio Reporting Summary linked to this article.

## Data availability
All data supporting the findings of this study are available in the main text or Supplementary Information. The accession numbers used in this study are LOC_Os01g05600; LOC_Os01g05620. Correspondence and requests for further data or materials should be addressed to Professor Yoji Kawano. Source data are provided in this paper.

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

## Acknowledgements

We thank the members of the Core Facility of Transformation and the Laboratory of Signal Transduction and Immunity at PSC for their valuable support and Dr. Keiko Hayashi (NARO) and Dr. Motoyuki Hattori (Fudan University) for their helpful discussion. Y. K. was supported by the Chinese Academy of Sciences, Shanghai Institutes for Biological Sciences, Shanghai Center for Plant Stress Biology, CAS Center of Excellence for Molecular Plant Sciences, Strategic Priority Research Program of the Chinese Academy of Sciences (B) (XDB27040202), the Chinese Academy of Sciences Hundred Talents Program (173176001000162114), the National Natural Science Foundation of China (31572073 and 31772246), JSPS KAKENHI (20H02988, 21H05035, 23H02213, 23K18030), JSPS Bilateral Programs (JPJSBP20237408), Ohara foundation, the Yakumo Foundation for Environmental Science, the Ryobi Teien Memory Foundation, the NOVARTIS Foundation (Japan) for the Promotion of Science, the Naito foundation, and the Joint Usage/Research Center, the Institute of Plant Science and Resources. K. K. was supported by the CAS President's International Fellowship Initiative (2019PB0056). H. J. was supported by the National Natural Science Foundation of China (31901985) and Shanghai Super Postdoctoral Fellow.

## Author contributions

Y.L. and Y.K. designed the study; Y.L., Q.W., H.J., K.I., K-I.K., T.U., A.T., M.Y., Y.Y., E.S., Y.F., M.F., A.S., H.N., F.F., Q.N., and Y.K. performed experiments and analyzed data; Y.L., H.J., K-I.K., and Y.K. wrote the manuscript; D.M., T.K.-K., L.T., M.F., R.A.W., C.K., M.S., K.Y., R.T., K.S., and Y.K. gave technical support and conceptual advice.

## Competing interests

The authors declare no competing interests.

## Additional information

[1]Shenzhen Branch, Guangdong Laboratory of Lingnan Modern Agriculture, Key Laboratory of Synthetic Biology, Ministry of Agriculture and Rural Affairs, Agricultural Genomics Institute at Shenzhen, Chinese Academy of Agricultural Sciences, Shenzhen 518120, China. [2]Shanghai Center for Plant Stress Biology, CAS Center for Excellence in Molecular Plant Sciences, Chinese Academy of Sciences, Shanghai 201602, China. [3]College of Plant Protection, Yangzhou University, Yangzhou 225009, China. [4]College of Agronomy, Jiangxi Agricultural University, Nanchang 330045 Jiangxi, China. [5]College of Life Sciences, Ritsumeikan University, Kusatsu 525-8577, Japan. [6]Fruit Tree Research Center, Ehime Research Institute of Agriculture, Forestry and Fisheries, Ehime 791-0112, Japan. [7]Laboratory of Plant Molecular Genetics, Nara Institute of Science and Technology, Nara 630-0101, Japan. [8]Faculty of Science, Kyushu University, Fukuoka 819-0395, Japan. [9]Department of Bioinformatics, Ritsumeikan University, Shiga 525-8577, Japan. [10]YANMAR HOLDINGS Co., Ltd., Osaka 530-8311, Japan. [11]College of Pharmaceutical Sciences, Ritsumeikan University, Shiga 525-8577, Japan. [12]Graduate School of Engineering Science, Yokohama National University, Yokohama, Kanagawa 240-8501, Japan. [13]Arizona Genomics Institute, School of Plant Sciences, University of Arizona, Tucson, AZ, USA. [14]Institute of Plant Science and Resources, Okayama University, Okayama 710-0046, Japan. [15]Advanced Academy, Anhui Agricultural University, Research Centre for Biological Breeding Technology, Hefei, Anhui 230036, China. [16]Iwate Biotechnology Research Center, Iwate 024-0003, Japan. [17]Graduate School of Agriculture, Kyoto University, Kyoto 617-0001, Japan. [18]Kihara Institute for Biological Research, Yokohama City University, Yokohama, Kanagawa 244-0813, Japan. [19]These authors contributed equally: Yuying Li, Qiong Wang, Huimin Jia. ✉e-mail: yoji.kawano@okayama-u.ac.jp

