## [Peer Review File · Nature Communications]

REVIEWER COMMENTS

Reviewer #1 (Remarks to the Author):

The paper by Li et al reports a detailed characterization of a pair of rice NLR-type immune receptors. An interesting observation by the authors is that a pair of homologous NLR genes called Pit1 and Pit2 appears to follow distinctive evolutionary process after a gene-duplication event and as a consequence Pit2 has obtained the ability to suppress host cell death and OsRac1 activation mediated by Pit1. Pit1 is known to confer disease resistance to an avirulent rice blast fungal race. Furthermore, the authors determined three key amino acid residues in the NB-ARC domain of Pit2 that are important for the suppressor function of Pit2. Interestingly, a Pit2 variant carrying the Pit1-type of the three amino acid residues ("LCG") is able to confer cell death and disease resistance to the avirulent rice blast fungus in the absence of Pit1, suggesting that Pit2 has a capacity to recognize an avirulent effector from rice blast fungus. This notion is supported by the evolutionary analysis on Pit2 genes of a set of rice accessions.

Overall, the paper reads well and the mechanism how Pit2 antagonizes Pit1 are well examined. However, a difficulty to understand this paper is that the disease resistance phenotype of the pit2 crispr-mutant to rice blast fungus is inconsistent with the cell death and OsRac1 data. The authors speculated that the phenotype of pit2 mutant might be due to compensation from other Pit2 homologs present in rice. To clarify this, analysis of Crispr/Cas9-mediated KO of other Pit2 homologs might be needed.

The Pit2 "LCG" variant confers resistance to the avirulent race 007.0 as Pit1. However, it is unclear if Pit1 and the Pit2 "LCG" variant recognize the same AVR effector of the race 007.0. As the cognate effector is not isolated, a panel of fungal races with different avirulent spectrum might be utilized. If Pit1 and the Pit2 "LCG" variant exhibit the same resistance pattern to such a fungal panel, this result will strongly support the authors hypothesis (line 20 at pg. 20) that Pit2 acts a sensor for the Pit1-mediated immunity. Please consider this experiment.

Specific comments are listed below.

1) Title: Which function of Pit2 is antagonized by Pit1? It is not clear from the current manuscript whether Pit1 and Pit2 are truly antagonistic each other.

2) Lines14 and 16 at pg5: "lipidation"? as NLRs themselves do not modify lipids at the PM.

- 3) Line 18 at pg 7: what is distance between Pit1 and Pit2 on the rice genome? Please indicate.
- 4) Line 25 at pg 7: Co-IP does not tell if co-precipitated proteins directly interact. There might be a bridging protein in-between.
- 5) Line 24 at pg 9: Please examine the transcript level of the other Pit homologues to see if the RNAi construct has off-target. The Pit1-dependent cell death upon the RNAi could be due to the silencing of other Pit genes, which might explain the discrepancy between the Pit2 RNAi and the Crispr KO of Pit2.
- 6) Lines 6-7 at pg 10. Is the obtained Pit2 allele a null allele? It seems that this mutant is still able to produce a truncated Pit2, which potentially interferes with the Pit1 function, as the authors showed that the CC-domains of Pit1 and Pit2 can interact. Also did the authors confirm that other Pit genes are not mutated?
- 7) Line 7 at pg 15. What exactly does "This" refer to? The discussion around is not clear.
- 8) Lines 16-19 at pg 16. How about Tajima's D of Pit3-Pit7?, which will help to deduce the roles of Pit3-Pit7. Also it would be very useful for readers if the same analysis is applied to RGA4 and RGA5 (if not done in the past.) and mentioned the result in Discussion section.
- 9) Fig. 3C F: do these lines epitope-tagged? No information is provided in the Method section. How about protein levels of each constructs?
- 10) Fig. 3F: If Pit2 merely antagonizes the Pit1-conditioned immunity (not cell death) to rice blast fungus, one can expect that the pit2 Crispr-mutant exhibits an enhanced disease resistance to virulent (compatible) rice blast fungus. Did the authors examine the disease resistance phenotype of the pit2 crispr-mutant to any virulent race?
- 11) Fig. 4D, right panel: Based on the band intensity of H⁺-ATPase, the loaded amount of proteins is not very equal, especially for the Pit2 LCG variant. This might hinder to reach the stated conclusion.

12) It is not always clear which genotype/accession of rice were used in individual assays. For example, what is the genotype of the rice suspension cell culture? Also, which genotype/accession were used in Fig.4CD? For the latter, Pit1 or Pit2 null background appear to be suitable.

13) Statistics: Student's t-test seemed to be used throughout.

Student's t-test is inappropriate to apply to multi-sample comparisons. Please use other statistics instead. Please also clarify whether the number of replicates in figure legend indicate the number of independent experiments or technical replicates within one independent experiment.

Reviewer #2 (Remarks to the Author):

This paper shows how pit1 and pit2 interact in rice and reveals their role in resistance. It contains a herculean effort to test the roles of the two genes. The manuscript contains a very large number of laboratory experiments that all seem to be done well. However, I have a major concern regarding the controls. In most cases only GPF has been used, or Pit1 and Pit2 serve as each others control. However, true negative controls would be the inclusion of a distally related NLR. Even though most of the experiments seem sound, this is quite important to do, because I find the observation that all domains of the Pits interact with all other domains of the other pit and itself, extremely odd. To me this seems very unlikely. Literature reports very specific dimerisation sites in other NLRs or in fact most modular proteins. These results suggest an artifact, due to a general stickiness of these construct, or in fact heir tags.

The part on the molecular evolution is very interesting and places the potential findings in a broader perspective, which would make the story more interesting for a general audience. The results are presented rather superficially, and vaguely. E.g. what does a "signature of selection" mean? Is there an indication for balancing or for negative selection? Are there changes from one to the other? . It is quite hard to get the relevant information out of the extended data table. A paper that claims to study the evolution of a gene family, should i.m.o. contribute a properly formatted main figure to the findings related in selection pressure. Overall, I think the whole section of molecular evolution of Pit1 and Pit2 would need a thorough overhaul. See also my specific comments below.

Comments

Fig1A A more meaningful negative control would be Co-IP with any of the other 6 Pit genes. That would truly show specificity of Pit1 and Pit2

Fig1F n = 3. Does that mean only 3 leaves were infected?

p9|15 I think the sentence is wrong. Pit2 specific for Pit1?

EFig4 The fact that everything IPs with everything is very dubious. meaningful controls would be different domains from other NLRs, including those of the previously described Pit3-7 and unrelated ones.

Fig2D. what are the amounts 2β 4? 6?

p11|12 barely? where is the mcerulean picture?

Fig2G I have not been keeping up with standards on these kind of WB analyses lately. Do these statements need statistical back ups Repeats and quantification? One WB alone seems vague to me.

p14|20. The fact that homologs exist in different species, does not imply they are conserved. They could be under diversifying selection.

pg14 what does AA, BB or FF genome mean? Not clear to someone not working on rice

p14|26 asian species have both. I don't see that *O. barthhii* or *glumaepatula* have only pit2? A general reader like me, will also not know which species are asian or not. This section should be clarified.

p15|4 a change in amino acid does not prove gene conversion. Could be a similar mutation. There is ample software about to test for gene conversions (e.g. through datamonkey). In some cases full gene alignments could also already show it clearly.

p15|5 where does this rate come from?

p16|11, I don't understand what intermediate average pi means. Please explain.

p19|2 are under positive selection

p20|22, lack the Pit1 gene, all of the accessions

p20|24, that they contribute to rice domestication is an overstatement

ExtDtable 3, for completeness and comparison, I would like to see Pit1 being split in NBS and LRR too. Is the C terminus included in the NBS or not analyzed?

Regarding to this, the methods do not indicate if how the domains were annotated and at which locations the genes were split for this analysis.

Why are the values for *O. rufipogon* for Tajima's D very negative?

The table needs reformatting to have the - sign with all numbers, it's now very confusing.

p16|20, the absolute value of Ka/Ks becomes more informative if the averages over the gene or the surrounding domains are given.

These kind of data would be required to interpret whether e.g. the overall gene is under balancing selection, but these sites are under purifying selection, because of their requirement for the interaction.

It seems to me that that is the kind of argument that the authors intend to make, but it is just not very clear to me.

p16l23 what do you mean with signature of selection? Positive? Balancing? Why do you highlight this for Pit2 and not for Pit1, where D is also negative?

p16l24 still observed in *O sativa*? values range from -1.7 in *rufipogon* to 2.6 in *sativa*. These suggest completely different evolutionary scenarios

p27l16, many of the graphs contain multiple bars, e.g. multiple means and thus multiple t-tests will have been done. In all these cases a multiple test correction needs to be performed (e.g. bonferroni).

The methods on generating the phylogenies and evolutionary analyses are not detailed enough. None of the parameters for the reconstruction are mentioned and at some point the methods section mentions that gaps and missing data were "processed". Without clear explanation on how these were processed the analyses cannot be redone. It makes a big difference if some variable sites are removed or not, so the authors should really elaborate on their methods. NJ trees are generally less reliable than ML trees. For complex genes like the NLR, I would resort to ML trees with clear definitions. I also think that the authors should provide all alignment files either as supplements or on a data repository, so people can assess the data themselves.

Finally, I think the title is a bit misleading. With a title like this I expect a paper with strong emphasis on the evolutionary trajectory of the Pit 1 and Pit2 genes, but in fact the paper is dominated by molecular biology analyses aiming to understand the function of the genes.

MS ID#: NCOMMS-20-16774

MS TITLE: Tandem gene duplication and evolution generate two antagonistic NLRs that confer resistance to rice blast fungus

We are grateful to the editor and both the reviewers for the critical comments and valuable suggestions that have helped us to improve our manuscript.

I relocated from China to Japan in 2020, just before the worldwide COVID-19 outbreak took place. Since then, I have been unable to return to China for an extended period due to the Zero Corona Policy implemented by the Chinese Government. Consequently, I was unable to access certain essential research materials that I left behind in China. These materials are crucial for conducting new experiments and addressing your comments. As a result, our revision process has taken considerably longer than usual. I kindly request your understanding and consideration of our extraordinary circumstances.

The revision has been made as follows according to the reviewers' comments.

REVIEWER COMMENTS

Reviewer #1 (Remarks to the Author):

The paper by Li et al reports a detailed characterization of a pair of rice NLR-type immune receptors. An interesting observation by the authors is that a pair of homologous NLR genes called Pit1 and Pit2 appears to follow distinctive evolutionary process after a gene-duplication event and as a consequence Pit2 has obtained the ability to suppress host cell death and OsRac1 activation mediated by Pit1. Pit1 is known to confer disease resistance to an avirulent rice blast fungal race. Furthermore, the authors determined three key amino acid residues in the NB-ARC domain of Pit2 that are important for the suppressor function of Pit2. Interestingly, a Pit2 variant carrying the Pit1-type of the three amino acid residues ("LCG") is able to confer cell death and disease resistance to the avirulent rice blast fungus in the absence of Pit1, suggesting that Pit2 has a capacity to recognize an avirulent effector from rice blast fungus. This notion is supported by the evolutionary analysis on Pit2 genes of a set of rice accessions.

1) Overall, the paper reads well and the mechanism how Pit2 antagonizes Pit1 are well examined. However, a difficulty to understand this paper is that the disease resistance phenotype of the pit2 crispr-mutant to rice blast fungus is inconsistent with the cell death and OsRac1 data. The authors speculated that the phenotype of pit2 mutant might be due to compensation from other Pit2 homologs present in rice. To

clarify this, analysis of Crispr/Cas9-mediated KO of other Pit2 homologs might be needed.

Response: Thank you for your comment. I apologize for the confusion caused by our vague description. The amino acid identity between Pit3–7 and Pit1 is low (Extended data Table 1), and Pit3–7 are not Pit2 homologs. To prevent this misleading, we have designated Pit3–7 as Pit1-associated NLR1–5 (PAN1-5) in the revised version.

We found that the transient knockdown of *Pit2* enhanced the expression of *PAN3* (Pit5) in K59 rice suspension cells, implying that *Pit2* is genetically linked to *PAN3* and *PAN3* also contributes to Pit1-mediated immunity (Extended Data Fig. 3F, page 10, lines 26-28). To investigate this possibility, we conducted the cell death assay using the *PAN3* knockdown K59 suspension cells. The transient knockdown of *PAN3* resulted in cell death mediated by Pit1, suggesting that *PAN3* functions as a suppressor to Pit1-induced cell death (Extended Data Fig.3G, page 10, lines 30-32). This result raises the possibility that *Pit2* KO leads to the increment of *PAN3* expression, resulting in the compensation of *Pit2* KO phenotype.

The rice blast resistance genes, *RGA4* and *RGA5*, function as a pair of NLRs. In *N. benthamiana* and rice suspension cells, the sensor NLR *RGA5* antagonizes the helper NLR *RGA4* (Cesari et al., EMBO J 2014: Fig. 1). Okuyama et al. discovered that when utilizing the rice cultivar Kanto 51, which lacks the endogenous *RGA4* and *RGA5* genes, the expression of the helper *RGA4* alone in Kanto 51 does not enhance disease resistance against the rice blast fungus. Instead, it requires the presence of the helper *RGA5* for the recognition of the effector and immune induction (Okuyama et al., Plant J 2011, Fig. 8b). These findings regarding *RGA4* and *RGA5* are consistent with those observed for Pit1 and Pit2. Mounting evidence indicates that sub-functionalized NLRs form a network-type receptor system called the NLR network (Wu et al., Proc Natl Acad Sci U S A 2017; Kourelis and Adachi, Plant Cell Physiol 2022). In this network, a limited subgroup of helper NLRs serves as a signaling hub for numerous sensor NLRs. It is possible that the helper Pit1 also forms an NLR network composed of Pit2 and PAN1–5, which compensate for the phenotype of the Pit2 KO. We have added this discussion in the revised version (page 17, lines 25-34 in the revised manuscript with track changes).

We agree with you that we did not provide sufficient data showing that Pit2 antagonizes Pit1 in planta. Therefore, we have toned down the statement of the antagonistic function of Pit2 against Pit1 throughout the manuscript (page 4, lines 10; page 17, lines 16 in the revised manuscript with track changes). Accordingly, we have changed the title to "Tandem gene duplication generates an NLR paralog that forms an NLR pair to confer resistance to rice blast fungus."

2) The Pit2 “LCG” variant confers resistance to the avirulent race 007.0 as Pit1. However, it is unclear if Pit1 and the Pit2 “LCG” variant recognize the same AVR effector of the race 007.0. As the cognate effector is not isolated, a panel of fungal races with different

avirulent spectrum might be utilized. If Pit1 and the Pit2 “LCG” variant exhibit the same resistance pattern to such a fungal panel, this result will strongly support the authors hypothesis (line 20 at pg. 20) that Pit2 acts a sensor for the Pit1-mediated immunity. Please consider this experiment.

Response: Thank you for your comment. As per your suggestion, we performed the infection assay using a different avirulent rice blast fungus Ina168 (Race 101.1) which also contains the potential cognate effector and displays a different avirulent spectrum. Pit1 WT and the Pit2 LCG variant exhibit the same resistance pattern as Ina168 (Extended Data Fig.9D and 9E page 14, lines 2 in the revised manuscript with track changes).

In collaboration with Dr. Ryohei Terauchi (Kyoto University), we have recently identified the effector for Pit1 and Pit2 named AvrPit (Sebastian, Shimizu, Yasuda et al., manuscript in preparation). We generated complementation lines using our AvrPit candidate in the virulent race of *M. oryzae* IB14-1k-1. Introducing *AvrPit* conferred a resistance phenotype in K59, which carries functional *Pit1* and *Pit2* genes, but not in K59 *Pit1* KO (Supplemental Fig. 1A for reviewer 1). The overexpression of the AvrPit mutant which lacks the N-terminal signal sequence (AvrPit Δ SS) induced cell death in K59, but not in the *Pit1* knockout of K59. These findings indicate that AvrPit functions as an authentic effector for Pit1 and Pit2 (Supplemental Fig. 1B for reviewer 1). Interestingly, AvrPit interacts with the LRR domain of Pit2, but not with that of Pit1 in the yeast two-hybrid assay (Supplemental Fig. 1C for reviewer 1). Since the LCG mutations are located in the NB-ARC domain of Pit2, these mutations are unlikely to affect effector recognition. These data support our hypothesis that Pit2 serves as a sensor for the effector AvrPit.

Specific comments are listed below.

3) Title: Which function of Pit2 is antagonized by Pit1? It is not clear from the current manuscript whether Pit1 and Pit2 are truly antagonistic each other.

Response: Thank you for your comment. As we discussed in comment 1 of reviewer 1, we agree that we did not provide sufficient data showing that Pit2 antagonizes Pit1 in planta. Therefore, we have toned down the statement of the antagonistic function of Pit2 against Pit1 throughout the manuscript (page 4, lines 10 and page 17, lines 16 in the revised manuscript with track changes). Accordingly, we have revised the title to "Tandem gene duplication generates a paralog that forms an NLR pair to confer resistance to rice blast fungus."

4) Lines14 and 16 at pg5: “lipidation”? as NLRs themselves do not modify lipids at the PM.

Response: Thank you for your comment. We have changed the sentence to “Some NLR proteins such as RPS5, RPS2, and RPM1 anchor themselves either directly or indirectly to

the plasma membrane through lipidation or by binding to a plasma membrane-localized guard protein known as RIN4" (page 6, lines 14-17 in the revised manuscript with track changes).

5) Line 18 at pg7: what is distance between Pit1 and Pit2 on the rice genome? Please indicate.

Response: Thank you for your comment. The distance between Pit1 and Pit2 is 9 kbp. We have added this information to the revised manuscript (page 8, lines 19 in the revised manuscript with track changes).

6) Line 25 at pg7: Co-IP does not tell if co-precipitated proteins directly interact. There might be a bridging protein in-between.

Response: Thank you for your comment. We have changed the sentence to "These results raise the possibility that Pit2 forms a complex with Pit1" (page 8, lines 26 in the revised manuscript with track changes).

7) Line 24 at pg9: Please examine the transcript level of the other Pit homologues to see if the RNAi construct has off-target. The Pit1-dependent cell death upon the RNAi could be due to the silencing of other Pit genes, which might explain the discrepancy between the Pit2 RNAi and the Crispr KO of Pit2.

Response: Thank you for your comment. As described in comment 1 for reviewer 1, we have examined the expression levels of *Pit1*, *Pit2*, and *PAN1–5* in *Pit2* RNAi K59 suspension cells and found that the transient knockdown of Pit2 enhanced the expression of PAN3 (Pit5) in K59 (Extended Data Fig. 3F: page 10, lines 26-29 in the revised manuscript with track changes).

8) Lines 6-7 at pg10. Is the obtained Pit2 allele a null allele? It seems that this mutant is still able to produce a truncated Pit2, which potentially interferes with the Pit1 function, as the authors showed that the CC-domains of Pit1 and Pit2 can interact. Also did the authors confirm that other Pit genes are not mutated?

Response: Thank you for your comment. We have generated *Pit2* KO plants, which are one base pair deletion at the position of 132, resulting in the production of a 78-aa long truncated Pit2 (Extended Data Fig. 4B). We tested whether this truncated Pit2 interacts with the CC domain Pit1 in a yeast-two hybrid assay and found that it was not associated with Pit1 (Supplemental Fig. 2A for reviewer 1). We have also co-expressed Pit1 with the truncated Pit2 in the rice protoplast system and checked cell death activity, and found that the truncated Pit2 did not suppress Pit1-induced cell death (Supplemental Fig. 2B for reviewer 1). We have

confirmed that the *Pit1* region corresponding to the target site of the *Pit2* knockout is intact by sanger sequencing (Extended Data Fig. 4B). Due to the low sequence homology between *Pit2* and *PAN1–5* (Table 1), we could not find any PAM sequences in *PAN1–5* that align with the target site of *Pit2*.

9) Line 7 at pg15. What exactly does "This" refer to? The discussion around is not clear.

Response: Thank you for your comment. As we described in comment 13 for reviewer 2, this sentence is not appropriate, we have deleted it in the revised manuscript. (page 15, lines 13-20 in the revised manuscript with track changes).

10) Lines 16-19 at pg 16. How about Tajima's D of Pit3-Pit7?, which will help to deduce the roles of Pit3-Pit7. Also it would be very useful for readers if the same analysis is applied to RGA4 and RGA5 (if not done in the past.) and mentioned the result in Discussion section.

Response: Thank you for your comment. In this manuscript, we wanted to focus on the analysis of Pit1 and Pit2. Consequently, we wanted to investigate the evolutionary process of *PAN1-5* in future studies.

A similar analysis was previously conducted on a pair of NLRs, RGA4, and RGA5 (Table 1: Okuyama et al., Plant J 2011). Notably, both the helper RGA4 and the sensor RGA5 exhibited a significant positive value for Tajima's D and Fu and Li's D. This indicates a clear signature of balancing selection imposed on RGA4 and RGA5. The selection pressure on sensors and helpers appears to vary among each NLR pair. We have added these sentences in the revised manuscript (page 16, lines 32-36 in the revised manuscript with track changes).

11) Fig.3C F: do these lines epitope-tagged? No information is provided in the Method section. How about protein levels of each constructs?

Response: Thank you for your comment. We have described the vector information we used in Fig 3C in Method section (page 22, lines 16-17 in the revised manuscript with track changes). We previously generated the Nipponbare expressing Pit1-HA; however, Pit1-HA was not fully functional in the infection assay (unpublished result). Furthermore, cell death induced by Pit1-Venus was significantly weaker compared to that induced by the non-tagged Pit1 in *N. benthamiana*. Therefore, we utilized the non-tagged Pit1 and Pit2 in Fig. 3C and 3F. To detect the protein level of Pit1, we produced an anti-Pit1 antibody; unfortunately, it failed to detect Pit1 WT in rice plants likely due to its low sensitivity. Consequently, it is technically difficult to determine the protein levels of Pit1 and Pit2 in Fig. 3C and 3F. However, the protein levels among Pit1 WT, Pit2 WT, and Pit2 LCG were comparable in *N. benthamiana* (Supplementary Fig. 3 for reviewer 1). Thus, it is unlikely that the infection result reflects the protein levels of Pit1 WT, Pit2 WT, and Pit2 LCG.

12) Fig.3F: If Pit2 merely antagonizes the Pit1-conditioned immunity (not cell death) to rice blast fungus, one can expect that the pit2 Crispr-mutant exhibits an enhanced disease resistance to virulent (compatible) rice blast fungus. Did the authors examine the disease resistance phenotype of the pit2 crispr-mutant to any virulent race?

Response: Thank you for your comment. As per your suggestion, we have performed the infection assay using virulent rice blast fungus. The lesion induced by *M. oryzae* in the *Pit2* KO was comparable to that in the WT (Supplementary Fig. 4 for reviewer 1).

13) Fig. 4D, right panel: Based on the band intensity of H⁺-ATPase, the loaded amount of proteins is not very equal, especially for the Pit2 LCG variant. This might hinder to reach the stated conclusion.

Response: Thank you for your comment. We have carefully checked the other experiments in Fig. 4D and discovered compelling evidence indicating that, while the bands of H⁺-ATPase between Pit2 WT and Pit2 LCG appeared comparable, a significantly higher accumulation of Pit2 LCG was observed in the membrane fraction compared to Pit2 WT. We have substituted the original Fig. 4D with this new data in the revised manuscript.

14) It is not always clear which genotype/accession of rice were used in individual assays. For example, what is the genotype of the rice suspension cell culture? Also, which genotype/accession were used in Fig.4CD? For the latter, Pit1 or Pit2 null background appear to be suitable.

Response: Thank you for your comment. We have specified the genotype and accession of rice throughout the manuscript. In Fig. 4C, we have used the *Oryza sativa indica* Oc cells which possess *Pit2* but disrupt *Pit1* function (page 14, lines 13-14 in the revised manuscript with track changes). In Fig. 4D, we have employed *N. benthamiana* to conduct subcellular fractionation.

15) Statistics: Student's t-test seemed to be used throughout. Student's t-test is inappropriate to apply to multi-sample comparisons. Please use other statistics instead. Please also clarify whether the number of replicates in figure legend indicate the number of independent experiments or technical replicates within one independent experiment.

Response: Thank you for your comment. We have performed one-way ANOVA for multi-sample comparisons and obtained consistent results throughout the figures. We have described the number of independent experiments or technical replicates within one independent experiment in the figure legends.

Other changes

16) We have corrected the estimated duplication time of *Pit* gene in the revised version (Fig 5A, page 15, line 17).

17) We have added several sentences in Introduction to cite important papers that have been published after the submission of this manuscript.

Reviewer #2:

1) This paper shows how pit1 and pit2 interact in rice and reveals their role in resistance. It contains a herculean effort to test the roles of the two genes. The manuscript contains a very large number of laboratory experiments that all seem to be done well. However, I have a major concern regarding the controls. In most cases only GFP has been used, or Pit1 and Pit2 serve as each others control. However, true negative controls would be the inclusion of a distally related NLR. Even though most of the experiments seem sound, this is quite important to do, because I find the observation that all domains of the Pits interact with all other domains of the other pit and itself, extremely odd. To me this seems very unlikely. Literature reports very specific dimerisation sites in other NLRs or in fact most modular proteins. These results suggest an artifact, due to a general stickiness of these construct, or in fact heir tags.

Response: We are grateful for your critical comments and useful suggestions, which have helped us improve our manuscript and figures.

To test the interaction between Pit1 and Pit2, we employed three different approaches: 1) Pit2-Myc coprecipitated with Pit1-HA but not with another rice NLR protein, RGA4, in *N. benthamiana* (Fig. 1A). 2) The consistent results were obtained in the transient assay using rice protoplast (Extended Data Fig. 1B). 3) We have generated transgenic rice plants carrying Pit1-Flag and Pit2-Myc driven by their native promoters, and found due to the co-IP assay that Pit1 interacted with Pit2 in rice (Extended Data Fig. 1C). Taken together, our results indicate that Pit1 indeed associates with Pit2 in rice.

Responding to the second concern raised by reviewer 2, we have addressed the heterodimer interaction of Pit1 and Pit2 by employing deletion mutants of RGA4 as negative controls (Supplementary Fig. 1 for reviewer 2). Under the condition that none of the RGA4 domains are associated with the corresponding domains of Pit1 and Pit2, all domains of Pit1 and Pit2 interacted with their respective counterparts and with each other. In line with the three-dimensional structure of the NLR protein ZAR1 pentamer, the CC and NB-ARC domains of ZAR1 form homointeractions (Wang et al., Science 2019). This result is consistent with those obtained in Pit1 and Pit2 (Supplementary Fig. 1 for reviewer 2). However, it is worth noting that the individual LRR domains of ZAR1 appear to be physically separated. As pointed out by reviewer 2, while the self- and intermolecular interactions of the domains of Pit1 and Pit2 are quite essential, it is technically difficult to obtain conclusive data without a three-dimensional structural analysis. Furthermore, the self- and intermolecular interactions of the CC and NB-ARC domains of Pit1 and Pit2 exceed the scope of our main hypothesis in this manuscript. Therefore, we have omitted these data in the revised version.

2) The part on the molecular evolution is very interesting and places the potential

findings in a broader perspective, which would make the story more interesting for a general audience. The results are presented rather superficially, and vaguely. E.g. what does a "signature of selection" mean? Is there an indication for balancing or for negative selection? Are there changes from one to the other? . It is quite hard to get the relevant information out of the extended data table. A paper that claims to study the evolution of a gene family, should i.m.o. contribute a properly formatted main figure to the findings related in selection pressure. Overall, I think the whole section of molecular evolution of *Pit1* and *Pit2* would need a thorough overhaul. See also my specific comments below.

Response) Thank you for your valuable comments. We have overhauled the whole section of the molecular evolution of *Pit1* and *Pit2* based on your instructions. Could you please see our responses below?

Comments

3) Fig1A A more meaningful negative control would be Co-IP with any of the other 6 Pit genes. That would truly show specificity of Pit1 and Pit2

Response: Thank you for your comment. As described in comment 1 of reviewer 2, to test the interaction between *Pit1* and *Pit2*, we performed co-IP assays using another rice NLR protein, RGA4, as a negative control in *N. benthamiana* leaves (Fig. 1A) and rice protoplasts (Extended Data Fig. 1B) and found that *Pit2* coprecipitated with *Pit1* but not with RGA4 in both experiments. Moreover, we generated transgenic rice plants expressing *Pit1*-Flag and *Pit2*-Myc driven by their respective native promoters (Extended Data Fig. 1C). Subsequent co-IP assays confirmed the interaction of *Pit1* and *Pit2* in rice. Collectively, these results provide evidence that *Pit1* indeed interacts with *Pit2* in rice.

We found that the transient knockdown of *Pit2* enhanced the expression of *PAN3* (*Pit5*) in K59 rice suspension cells, implying that *Pit2* is genetically linked to *PAN3* and *PAN3* also contributes to *Pit1*-mediated immunity (Extended Data Fig. 3F, page 10, lines 26-28). To investigate this possibility, we conducted the cell death assay using the *PAN3* knockdown K59 suspension cells. The transient knockdown of *PAN3* resulted in cell death mediated by *Pit1*, suggesting that *PAN3* also functions as a suppressor to *Pit1*-induced cell death (Extended Data Fig. 3G, page 10, lines 29-32). Mounting evidence suggests that sub-functionalized NLRs participate in a network-type receptor system referred to as the NLR network. This network relies on the cooperation of more than three NLRs, cooperating to recognize diverse pathogen effectors and activate immune signaling (Wu et al., Proc Natl Acad Sci U S A. 2017; Kourelis and Adachi, Plant Cell Physiol. 2022). One helper NLR forms a complex network(s) with multiple sensor NLRs. Based on this notion, we infer that *Pit2* is not the sole binding partner of *Pit1* and that *Pit1* forms a network-type receptor system with other NLRs. We have added these sentences in the revised version (page 18, lines 11-14).

Please note that the amino acid identity between Pit3–7 and Pit1 is low (Table 1), and Pit3–7 are not Pit1 homologs. In order to avoid potential confusion and misinterpretation of Pit3–7 as Pit1 homologs, we have designated Pit3–7 as Pit1-associated NLR1–5 (PAN1–5) in the revised version (Table 1).

4) Fig1F n = 3. Does that mean only 3 leaves were infected?

Response:

Thank you for your comment. Fig. 1F shows the cell death assay performed on rice suspension cells. Three biological samples were prepared for each experiment, and the average value of these three samples was calculated. The experiment was repeated two additional times, and the statistical analysis using the average value of the three experiments was conducted as depicted in Fig. 1F. We have described these sentences in the revised manuscript (page 34; Lines 31-33).

5) p9115 I think the sentence is wrong. Pit2 specific for Pit1?

Response: Thank you for your comment. We agree with your opinion and have deleted this sentence.

6) EFig4 The fact that everything IPs with everything is very dubious. meaningful controls would be different domains from other NLRs, including those of the previously described Pit3-7 and unrelated ones.

Response: Thank you for your comment. We have examined the homooligomer and heterooligomer formation of the Pit1 and Pit2 deletion mutants using the corresponding RGA4 deletion mutants as negative controls (Supplementary Fig. 2 for reviewer 2). Our data showed that the CC, NB-ARC, and LRR domains of Pit1 and Pit2 form homooligomers and heterooligomers, but did not interact with those of RGA4. As described in comment 1 of reviewer 2, it is technically difficult to obtain conclusive data without a three-dimensional structural analysis. Therefore, we have omitted these data in the revised version.

7) Fig2D. what are the amounts 2B 4? 6?

Response: Thank you for your comment. The values 2, 4, and 6 correspond to the protein quantity of 200, 400, and 600 pmol of Pit2-CC-Myc in the in vitro binding assay samples. We have added this detailed information in Fig. 2D of the revised manuscript (page 26, lines 7-8 in the revised manuscript with track changes).

8) p11112 barely? where is the mcerulean picture?

Response: Thank you for your comment. We have revised this sentence to “Pit2-Venus “hardly” merged with mCerulean fused to a nuclear import signal.” We have added the mcerulean data in the revised version (Fig 2E).

9) Fig2G I have not been keeping up with standards on these kind of WB analyses lately. Do these statements need statistical back ups Repeats and quantification? One WB alone seems vague to me.

Response: Thank you for your comment. After analyzing the band intensity of Pit1, we conducted a statistical analysis and obtained consistent results in our conclusion. We have provided a more comprehensive description of the experimental conditions in the Figure legends in the revised manuscript (Extended Data Fig. 5E, page 41, lines 7 in the revised manuscript with track changes).

10) p14I20. The fact that homologs exist in different species, does not imply they are conserved. They could be under diversifying selection.

Response: Thank you for your comment. We apologize for the confusion caused by our vague description. As described in comment 2 of reviewer 2, the amino acid identity between Pit3–7 and Pit1 is low (Table 1), and Pit3–7 are not Pit1 homologs. To prevent misleading information, we designated Pit3–7 as Pit1-associated-NLR1-5 (PAN1-5) in the revised version (page 8, lines 14 in the revised manuscript with track changes). We conducted a thorough search for Pit1 homologs in both wild and cultivated rice, as illustrated in Fig. 5B. but were unable to identify any Pit1 homologs apart from Pit2. These findings do not provide evidence in favor of diversifying selection.

11) pg14 what does AA, BB or FF genome mean? Not clear to someone not working on rice

Response: Thank you for your comment. AA, BB, and FF are the genome symbols of rice. The genome symbol from A to F is assigned to each rice species based on chromosome pairing during meiosis in the first generation of hybrid germplasm. Generally, there are no abnormalities in chromosome pairing in the hybrid of species with the same symbols. We described this information in the revised version (page 15, lines 1-3 in the revised manuscript with track changes).

12) p14I26 asian species have both. I don't see that *O. barthii* or *glumaepatula* have only pit2? A general reader like me, will also not know which species are asian or not. This section should be clarified.

Response: Thank you for your comment. We apologize for the ambiguous description and incorrect species labeling. We have revised the sentence as follows: "*Oryza punctata* (BB

genome) has a *Pit2* gene, and all Asian *Oryza* species including *O. nivara*, *O. sativa* *vg. india*, and *O. sativa* *vg. japonica* (AA genome), except for *O. rufipogon*, have both *Pit1* and *Pit2* genes (page 15, lines 5-7).

13) p1514 a change in amino acid does not prove gene conversion. Could be a similar mutation. There is ample software about to test for gene conversions (e.g. through datamonkey). In some cases full gene alignments could also already show it clearly.

Response: Thank you for your comment. To address your suggestion, we conducted a full gene alignment of the *Pit2* (KN542601.1) of *O. longistaminata* with the other *Pit1* and *Pit2* genes (Supplemental Fig. 3 for reviewer 2). We highlighted the nucleotides with the red boxes where KN542601.1 matches with the *Pit-2* genes. KN542601.1 has several SNPs where *Pit-1* and *Pit-2* are nested. It cannot explain a simple single-gene conversion between *Pit1* and *Pit2*. Therefore, we have toned down the statement on the gene conversion between *Pit1* and *Pit2* in *O. longistaminata*. Accordingly, we have revised the sentences to "*Pit2* (KN542601.1) of *O. longistaminata* clustered into a *Pit1* clade, which is likely due to the gene conversion between *Pit1* and *Pit2* (page 15, lines 13-16 in the revised manuscript with track changes).

14) p1515 where does this rate come from?

Response: Thank you for your comment. We have employed the formula $T = Ks/2\lambda$ to calculate the duplication and divergence time. λ referred to the mutation rate and was considered as 6.5×10^{-9} synonymous substitutions per site per year (Molina et al., PNAS 2011). We have added this sentence in the revised version (page 15, lines 14-16 in the revised manuscript with track changes)

Molina J, Sikora M, Garud N, Flowers JM, Rubinstein S, Reynolds A, Huang P, Jackson S, Schaal BA, Bustamante CD et al. 2011. Molecular evidence for a single evolutionary origin of domesticated rice. *Proceedings of the National Academy of Sciences* **108**: 8351-8356.

15) p16111, I dont understand what intermediate average pi means. Please explain.

Response: Thank you for your comment. Yang et al. previously analyzed the average nucleotide diversity (π) of 44 *NLR* genes among 21 rice cultivars and 14 wild rice populations. They categorized the genes into four distinct groups: 1) conserved ($\pi < 0.005$), 2) diversified ($\pi > 0.05$), 3) intermediate-diversified ($\pi = 0.005-0.05$), and 4) present/absent patterns (Yang et al, 2006). The average π value for *Pit1* alleles in *O. sativa* was 0.01585, while for *Pit2* alleles, it was 0.00349. This result suggests that the *Pit1* gene exhibits intermediate levels of nucleotide variation, whereas the *Pit2* gene shows low nucleotide diversity. (Extended Data Table 3, page 16, lines 17-25 in the revised manuscript with track changes).

Yang S, Feng Z, Zhang X, Jiang K, Jin X, Hang Y, Chen JQ, Tian D (2006) Genome-wide investigation on the genetic variations of rice disease resistance genes. *Plant Mol Biol* 62:181–193

16) p19I2 are under positive selection

Response: Thank you for your comment. We have changed the word in the revised manuscript (page 19, lines 16 in the revised manuscript with track changes).

17) p20I22, lack the Pit1 gene, all of the accessions

Response: Thank you for your comment. We have changed the word in the revised manuscript (page 21, lines 7 in the revised manuscript with track changes).

18) p20I24, that they contribute to rice domestication is an overstatement

Response: Thank you for your comment. We have toned down this point in the revised manuscript (page 21, lines 9 in the revised manuscript with track changes).

19) ExtDtable 3, for completeness and comparison, I would like to see Pit1 being split in NBS and LRR too. Is the C terminus included in the NBS or not analyzed? Regarded to this, the methods do not indicate if how the domains were annotated and at which locations the genes were split for this analysis. Why are the values for *O. rufipogon* for Tajima'S D very negative? The table needs reformatting to have the - sign with all numbers, it's now very confusing.

Response: Thank you for your comment. We have split Pit1 into the CC (1–140 aa), NB (141–520 aa), and LRR (521–989 aa) domains and examined their nucleotide polymorphism in the revised version of the extended Table 3.

We realized that we only utilized seven accessions of nucleotide polymorphism data for Pit1 and Pit2 alleles in *O. rufipogon* (Extended Data Table 2). This number is inadequate for this type of analysis, leading us to exclude the results of *O. rufipogon* in the revised manuscript.

We meticulously reformatted the content of Extended Data Table 3.

20) p16I20, the absolute value of Ka/ks becomes more informative if the averages over the gene or the surrounding domains are given. These kind off data would be required to interpret whether e.g. the overall gene is under balancing selection, but these sites are under purifying selection, because of their requirement for the interaction. It seems to me that that is the kind of argument that the authors intend to make, but it is just not very clear to me.

In the revised version, we have provided the absolute Ka/Ks value of the averages over the *Pit1* and *Pit2* genes (Extended Data Tables 5 and 6).

The Ka/Ks analysis on the three important residues of the functional difference between Pit1 and Pit2 and the selection server program on Pit2 revealed that P300 and F415 in Pit2 have positive selection (Extended Data Table 3 and Extended Data Fig. 13). The following three results support that the *Pit2* gene encoding the C-terminal LRR domain of Pit2 is under balancing selection. 1) The Tajima's *D* value in the LRR domain of Pit2 in *O. sativa* and *O. sativa japonica* is positive (Extended Data Table 3). 2) The ML tree of Pit2 divides into two major clusters (Extended Data Fig. 12). 3) The selection server program shows that Pit2 accumulates the positively selected residues at the C-terminal LRR domain of Pit2 (Extended Data Fig. 13). We have added this description in the revised version (page 20, lines 23-28).

21) p16l23 what do you mean with signature of selection? Positive? Balancing? Why do you highlight this for Pit2 and not for Pit1, where D is also negative?

Response: Thank you for your comment. As described in comment 19 for reviewer 2, the number of accessions of *O. rufipogon* is low, and we have excluded the results of *O. rufipogon* in the revised manuscript of Extended Data Table 3.

22) p16l24 still observed in O sativa? values range from -1.7 in rufipogon to 2.6 in sativa. These suggest completely different evolutionary scenarios

Response: Thank you for your comment. As described in comment 19 for reviewer 2, due to this limited sample size of *O. rufupogon*, we have excluded the data of *O. rufupogon* in the revised version of Extended Data Table 3. Consequently, we have also modified the sentence to state that " the LRR region of Pit2 in *O. oryza japonica* but not in *O. oryza indica* shows significant signature of positive selection" (Extended Data Fig. 13 and Extended Data Table 3; page 17, lines 5-6).

23) p27l16, many of the graphs contain multiple bars, e.g. multiple means and thus multiple t-tests will have been done. In all these cases a multiple test correction needs to be performed (e.g. bonferroni).

Response: We thank you for your suggestion. We have performed one-way ANOVA throughout the study.

24) The methods on generating the phylogenies and evolutionary analyses are not detailed enough. Noe of the parameters for the reconstruction are mentioned nd at some point the methods section mentions that gaps and missing data were "processed". Without clear explanation on how these were processed the analyses cannot be redone. It makes a big difference if some variable sites are removed or not,

so the authors should really elaborate on their methods. NJ trees are generally less reliable than ML trees. For complex genes like the NLR, I would resort to ML trees with clear definitions. I also think that the authors should provide all alignment files either as supplements or on a data repository, so people can assess the data themselves.

Response:

Response: Thank you for your comment. We have included a comprehensive explanation of the phylogenies' generation process and evolutionary analyses in the Materials and Methods section (Extended Data Fig. 12 and Extended Data Table 3). Additionally, we have generated the ML trees (Extended Data Fig. 12 and Extended Data Table 3) and provide the sequence data as supplemental materials (Extended Data Fig. 12 and Extended Data Table 3).

25) Finally, I think the title is a bit misleading. With a title like this I expect a paper with strong emphasis on the evolutionary trajectory of the *Pit1* and *Pit2* genes, but in fact the paper is dominated by molecular biology analyses aiming to understand the function of the genes.

Response: Thank you for your comment. We have toned down the statement of the evolutionary trajectory of the *Pit1* and *Pit2* genes. Accordingly, we have changed the title to "Tandem gene duplication generates a paralog that forms an NLR pair to confer resistance to rice blast fungus."

Other changes

16) We have corrected the estimated duplication time of *Pit* gene in the revised version (Fig 5A, page 15, line 17).

17) We have added several sentences in Introduction to cite important papers that have been published after the submission of this manuscript.

REVIEWER COMMENTS

Reviewer #1 (Remarks to the Author):

The authors' response and revisions have satisfactorily addressed my concerns.

However, I would suggest including some of the supplementary figures for reviewer 1 (e.g. Fig. 4 for reviewer 1).

Reviewer #3 (Remarks to the Author):

Review on Pit1-2

Yuying Li and colleagues present a massive body of work of reporting an NLR pair in rice with in depth functional characterization regarding their mode of action in cell death and disease resistance. This research had a privilege of expanding their initial finding of the rice R protein Pit1 interacting with other NLRs, owing to previously established research from the same group and collaborators. Detailed molecular analyses of how the newly identified paralog Pit2 acts with Pit1 were carried out to define their downstream mode of action, such as OsRac1, as well as subcellular localization. The authors nicely demonstrated a solid physical interaction between Pit1 and Pit2, followed by detailed characterization of the two NLR cooperation in terms of triggering (or fine-tuning) immune responses and localization differences. The strength of this work lies in the detailed characterization functional differences of the closely related two paralogous NLRs, provision of their evolutionary history, and delineation of the important residues and their contribution to the functional differences between the two. All these provides the explanation of how neofunctionalization in Pit2 had occurred on the duplicate copy of Pit1, apparently ancient than the derived Pit2. The evolutionary analyses appear sound and solid with molecular clock dating to reveal the origin of this duplicate within the *Oryza* lineage. Further comparison was made on the three key residues that correspond to the sub-clades of Pit1/Pit2 according to the tree provided, indicating their importance in shaping their evolutionary trajectory. Again, the amount of work and the degree of execution, in terms of provision of multiple lines of evidence for one notion, e.g. physical interaction, should be praised and appreciated as major achievements.

While data is massive and all individually solid, I reckon that the degree of synthesis of the finding can largely improve. It is still not clear whether or not the authors want to propose Pit2 as a sensor, which seem far stretched to my understanding. Here are the points that I would like to challenge to make the manuscript truly be written to reflect their findings.

1. The discovery of Pit2 as a tandem (claimed, see below specific comments) pair of Pit1 using the autoactive Pit1 MHV is quite a nice approach. However, innately this way of approach conditions that

Pit2 is tethered to an active signaling complex of Pit1, potentially suppressing overtly activating Pit1 NLR. With evolutionary study, the plausible scenario would be that Pit2 had evolved from a diversification of the tandem duplicate of Pit1 to control and counter the membrane-bound NLR function of Pit1, supposedly cell death execution per se, as illustrated in Figure 6. In this sense, the essence of Pit2 function lies in the fine-control of Pit1 cell death function not to overly activate immunity when needed. However, throughout the manuscript, this pair was implied and discussed in line to sensor-helper/executor function. Given the balancing selection signature of Pit2, low degree of diversification within the lineage (based on tree) and provided functional data, it is hard to imagine how Pit2 is seen as a sensor. For example, introduction primes the reader (P6, L8) of the helper-sensor dichotomy.

I would think that the authors may need to clarify the proposed function of Pit2 based on the data, rather antagonizing like PigmS/R. Current writing and discussion are not clear enough to grasp the proposed function of Pit2.

2. What is the Pit2 expression pattern upon avirulent pathogen infection and what would happen when Pit2 is overly expressed to the level of suppressing the action of Pit1? Would it compromise the Pit1-mediated disease resistance? Not asking for experiments per se, but these questions can be discussed in line with the proposed function of Pit2.

I would rather think that the provided evidence supports for the Pit2 acting as a fine-tuner of the Pit1 cell death executor, while the other PANs or unidentified NLRs function as sensors. Instead of fitting the data into the currently favored network of NLR and/or helper-sensor, the authors may resynthesize this fantastic data with a new angle.

3. Positive selection detected on the LRR in Pit2 shall be reconsidered. To properly assembly the horseshoe shape of LRR, there has to be a certain position to maintain Leucine (LXXL) with spacing residues showing a signature of diversifying selection for sensors in general. It would be good to tone down the assumption that positive selection signifies an NLR being as a sensor.

4. From the provided title, I could not see academic novelty. The model of action (quite well studied in this study) of Pit1/Pit2 is not well conveyed in the title. And it is not clear how much the duplicate copy Pit2 contributes to the disease resistance function of Pit1 (essential to form the heteromer to condition full resistance?)

5. P8, L19: If tandem duplication argument holds true, there needs to be an illustration of the locus as a main figure panel. Are there any other gene in the 9 kb and are these configured as head-to-head in their genomic placement? The locus information of Pit1 corresponds to LOC_Os01g05620 vs. Pit2 to LOC_Os01g05600. To general audience who is not familiar with genome annotation of rice, it would be difficult to see if there are placed next to each other in tandem.

I also list out minor comments that may help improve the manuscript when addressed.

1) It was claimed that PAN1-PAN5 are not orthologs of Pit1 or Pit2 and thus named differently. Is there any phylogenetic tree to support the less relatedness of PANs to Pits? The tree should include some other rice NLRs as outgroup.

2) ABSTRACT L11: Is the notion of “Pit1 acts as a helper of ETI” demonstrated? The presented data includes the HR conditioned by N. ben experiments using MHV/autoactive, but non-effector-triggered version. Is there any evident and literature backing up that Pit1 mediates effector-triggered immunity? Otherwise, this definitive sentence shall be reworded.

3) L17: no need to add comma after “a new function”.

4) In general, the gap of knowledge that this work addresses shall be much better articulated and novelty of this work shall be redefined. See the followings:

P5, L11: how duplicated NLR genes acquire new functions were rather very well studied in my opinion, e.g. L6/L7 and RPS4/RRS1 and its duplicate pair of RPS4B/RRS1B, alleles of NLRs causing autoimmunity as hybrid necrosis (Dangerous Mix genes). The reference 60 that they cited, for example, characterized allelic variation of the executor-sensor pair CHS3-CSA1 evolution. Authors shall clarify what had been known in the evolution of NLRs through duplication and sub/neo-functionalization.

P6, L8: Physical interaction between RPS4 and RRS1 as well as RGA4 and RGA5 were known, and there are other examples.

P6, L31-32: Compared to other NLRs, Pit1 downstream and signaling partners were identified with mode of action and localities. The sentence starting with “little is known” demotes their own finding. What authors meant would be that the mode of action of Pit1 NLR in relation to other NLRs are not known. Please articulate what is the main agenda of this research is here.

5) P7, L5: Unless identified, “the” ancestral Pit gene shall be referred to as “an” ancestral Pit gene.

6) (related to major points) PAN3 results in suppressing Pit1 indicates its participation to Pit1-triggered immunity. Are there any sequence homology of PAN3 (and other PANs) to Pit1 and Pit2? Are there any reliable phylogeny to be presented to demonstrate their relationship as compared to other rice NLRs?

7) P11, L9-11: It appears to be an overstatement to interpret the disease resistance data of Pit2 KO presented in Figure1G. To further demonstrate Pit2 participating the Pit1-mediate resistance, double mutant analyses might be required. Rewording shall be made on this notion if further data cannot be provided.

8) P16: CH3-CSA1 > CHS3-CSA1

9) NLR does not need to be italicized as the abbreviation itself represents protein domains. It is not the gene name, such as RPP1 or RPS2, of which abbreviation shall be italicized. “NLR (non-italics) genes” should be sufficient in delivering the messages of NLR-encoding genes.

MS ID#: NCOMMS-20-16774

MS TITLE: Tandem gene duplication and evolution generate two antagonistic NLRs that confer resistance to rice blast fungus

We are grateful to the editor and both reviewers for the critical comments and valuable suggestions that have helped us to improve our manuscript.

The revision has been made as follows according to the reviewers' comments.

REVIEWER COMMENTS

Reviewer #1

Reviewer #1 (Remarks to the Author):

The authors' response and revisions have satisfactorily addressed my concerns.

However, I would suggest including some of the supplementary figures for reviewer 1 (e.g. Fig. 4 for reviewer 1).

Thank you very much for being satisfied with our revised manuscript. As per your suggestion, we have put Supplemental Fig. 4 for Reviewer 1 as Extended Data Fig. 5D and 5E (page 12, lines 5-7; page 42, lines 15-20 in the revised manuscript with track changes).

Reviewer #3 (Remarks to the Author): Review on Pit1-2

Yuying Li and colleagues present a massive body of work of reporting an NLR pair in rice with in depth functional characterization regarding their mode of action in cell death and disease resistance. This research had a privilege of expanding their initial finding of the rice R protein Pit1 interacting with other NLRs, owing to previously established research from the same group and collaborators. Detailed molecular analyses of how the newly identified paralog Pit2 acts with Pit1 were carried out to define their downstream mode of action, such as OsRac1, as well as subcellular localization. The authors nicely demonstrated a solid physical interaction between Pit1 and Pit2, followed by detailed characterization of the two NLR cooperation in terms of triggering (or fine-tuning) immune responses and localization differences. The strength of this work lies in the detailed characterization functional differences of the closely related two paralogous NLRs, provision of their evolutionary history, and delineation of the important residues and their contribution to the functional differences between the two. All these provides the explanation of how neofunctionalization in Pit2 had occurred on the duplicate copy of Pit1, apparently ancient than the derived Pit2. The evolutionary analyses appear sound and solid with molecular clock dating to reveal the origin of this duplicate within the Oryza lineage. Further comparison was made on the three key residues that correspond to the sub-clades of Pit1/Pit2 according to the tree provided, indicating their importance in shaping their evolutionary trajectory. Again, the amount of work and the degree of execution, in terms of provision of multiple lines of evidence for one notion, e.g. physical interaction, should be praised and appreciated as major achievements.

1) While data is massive and all individually solid, I reckon that the degree of synthesis of the finding can largely improve. It is still not clear whether or not the authors want to propose Pit2 as a sensor, which seem far stretched to my understanding. Here are the points that I would like to challenge to make the manuscript truly be written to reflect their findings.

We agree that we did not provide sufficient data showing that Pit2 acts as a sensor NLR in planta in this manuscript. Therefore, we have toned down our statement on the sensor function of Pit2 throughout the manuscript (page 22, lines 28-page 23, line 3 in the revised manuscript with track changes).

We have recently acquired supporting data that Pit2 functions as an NLR sensor. In collaboration with Dr. Ryohei Terauchi from Kyoto University, we have identified an effector

candidate for Pit1 and Pit2, named AvrPit (Sebastian, Shimizu, Yasuda et al., unpublished data). Complementation lines were generated using our AvrPit candidate in the virulent race of *M. oryzae* IB14-1k-1. The introduction of AvrPit conferred a resistance phenotype in K59, which carries functional *Pit1* and *Pit2* genes, but not in *Pit1* KO of K59 (Supplemental Fig. 1A for reviewer 3). Overexpression of the AvrPit mutant lacking the N-terminal signal sequence (AvrPit Δ SS) induced cell death in K59 suspension cells, but not in *Pit1* KO of K59 (Supplemental Fig. 1B for reviewer 3). These findings suggest that AvrPit is an authentic effector for Pit1 and Pit2.

Furthermore, AvrPit interacted with the LRR domain of Pit2 but not with that of Pit1 in the yeast two-hybrid assay (Supplemental Fig. 1C and 1D for reviewer 3). Interestingly, the binding region of AvrPit on Pit2 is the C-terminal edge of the Pit2 LRR domain, aligning with the positive selection residues identified in Pit2 (Extended Data Fig. 14). These data support our hypothesis that Pit2 functions as a sensor for the effector AvrPit. While we have not included the AvrPit data in this manuscript, and acknowledge that the data presented here would not conclusively support our hypothesis, we wish to retain certain sentences and references about the sensor function of Pit2.

2) The discovery of Pit2 as a tandem (claimed, see below specific comments) pair of Pit1 using the autoactive Pit1 MHV is quite a nice approach. However, innately this way of approach conditions that Pit2 is tethered to an active signaling complex of Pit1, potentially suppressing overtly activating Pit1 NLR. With evolutionary study, the plausible scenario would be that Pit2 had evolved from a diversification of the tandem duplicate of Pit1 to control and counter the membrane-bound NLR function of Pit1, supposedly cell death execution per se, as illustrated in Figure 6. In this sense, the essence of Pit2 function lies in the fine-control of Pit1 cell death function not to overly activate immunity when needed. However, throughout the manuscript, this pair was implied and discussed in line to sensor-helper/executor function. Given the balancing selection signature of Pit2, low degree of diversification within the lineage (based on tree) and provided functional data, it is hard to imagine how Pit2 is seen as a sensor. For example, introduction primes the reader (P6, L8) of the helper-sensor dichotomy.

I would think that the authors may need to clarify the proposed function of Pit2 based on the data, rather antagonizing like PigmS/R. Current writing and discussion are not clear enough to grasp the proposed function of Pit2.

Thank you for your constructive comments. We have carefully reorganized the Introduction (page 5, line 21–page 7, line 8 in the revised manuscript with track changes) and Discussion sections

(page 18, line 1–page 18, line 18; page 19, lines 24–28; page 21 lines 23–page 22, line 4; page 22, lines 9–24; page 23, lines 6–8 in the revised manuscript with track changes) to explain our data. We concur with your opinion that the core significance of the Pit2 function elucidated in this manuscript lies in the fine control of the Pit1 cell death function. Therefore, we have revised our statement on the sensor function of Pit2 throughout the manuscript (page 22, line 28–page 23, line 3). We have shortened the introduction regarding the helper-sensor dichotomy (page 5, lines 28–31; page 6, lines 2–3, lines 5–8, lines 10–15 in the revised manuscript with track changes) and have clearly discussed the possibility that Pit2 has evolved from a diversification of the tandem duplication of Pit1 to control the membrane-bound Pit1 function to fine-tune Pit1 cell death execution in the revised version of the Discussion (page 21, line 23–page 22, line 4 in the revised manuscript with track changes). We have also toned down our statement on the balancing selection signature of Pit2 throughout the manuscript (page 17, line 12 and line 22; page 22 lines 7–9; page 22, lines 28–page 23, line 3; page 40, line 8 in the revised manuscript with track changes)

3) What is the Pi2 expression pattern upon avirulent pathogen infection and what would happen when Pit2 is overly expressed to the level of suppressing the action of Pit1? Would it compromise the Pit1-mediated disease resistance? Not asking for experiments per se, but these questions can be discussed in line with the proposed function of Pit2.

I would rather think that the provided evidence supports for the Pit2 acting as a fine-tuner of the Pit1 cell death executor, while the other PANs or unidentified NLRs function as sensors. Instead of fitting the data into the currently favored network of NLR and/or helper-sensor, the authors may resynthesize this fantastic data with a new angle.

We have not tested the *Pit2* expression upon an avirulent *M. oryzae* infection and the effect of *Pit2* overexpression on Pit1-mediated disease resistance. In the revised version of the manuscript, we have added new sentences to discuss the proposed function of Pit2 (page 18, lines 1–18; page 21, lines 30–34; page 21, line 23–page 22, line 4; page 22 lines 9–23; page 23, lines 6–8 in the revised manuscript with track changes).

As we described in our response to comment 1 of Reviewer 3, our latest data suggest that Pit2 acts as a sensor for AvrPit. Since the data presented here would not be sufficient to support our hypothesis, we have revised our statement on the sensor function of Pit2 throughout the manuscript and discussed the fine-tuning function of Pit2 in more detail (page 21, lines 34–page 22, line 4; pages 22, lines 9–23; page 23, lines 6–8 in the revised manuscript with track changes).

4) Positive selection detected on the LRR in Pit2 shall be reconsidered. To properly assemble the horseshoe shape of LRR, there has to be a certain position to maintain

Leucine (LXXL) with spacing residues showing a signature of diversifying selection for sensors in general. It would be good to tone down the assumption that positive selection signifies an NLR being as a sensor.

We have toned down our assumption that positive selection signifies an NLR as a sensor (page 22, lines 28-page 23, line 3 in the revised manuscript with track changes).

5) From the provided title, I could not see academic novelty. The model of action (quite well studied in this study) of Pit1/Pit2 is not well conveyed in the title. And it is not clear how much the duplicate copy Pit2 contributes to the disease resistance function of Pit1 (essential to form the heteromer to condition full resistance?)

We have changed the title to " An NLR paralog Pit2 generated from tandem duplication of *Pit1* fine-tunes Pit1 localization and function" (page 1, lines 3-4 in the revised manuscript with track changes).

6) P8, L19: If tandem duplication argument holds true, there needs to be an illustration of the locus as a main figure panel. Are there any other gene in the 9 kb and are these configured as head-to-head in their genomic placement? The locus information of Pit1 corresponds to LOC_Os01g05620 vs. Pit2 to LOC_Os01g05600. To general audience who is not familiar with genome annotation of rice, it would be difficult to see if there are placed next to each other in tandem.

In the revised version, we have added an illustration of the loci of *Pit1* and *Pit2* genes in Extended Data Fig. 1B (page 9, line 20; page 40, line 17 in the revised manuscript with track changes).

I also list out minor comments that may help improve the manuscript when addressed.

7) It was claimed that PAN1-PAN5 are not orthologs of Pit1 or Pit2 and thus named differently. Is there any phylogenetic tree to support the less relatedness of PANs to Pits? The tree should include some other rice NLRs as outgroup.

The amino acid identity between Pit1 and PAN1-5 is no more than 34% (Extended Data Table 1). Moreover, our phylogenetic tree analysis using 430 NLR protein sequences (Ding et al, 2020) also supports that PAN1-PAN5 are not orthologs of Pit1 (Extended Data Fig. 2; page 9, line 20; page 9, lines 22-23; page 40, line 31-page 41, line 2 in the revised manuscript with track changes).

Reference) Ding et al., Genome-wide identification and expression analysis of rice NLR genes responsive to the infections of *Xanthomonas oryzae* pv. *oryzae* and *Magnaporthe oryzae* Physiological and Molecular Plant Pathology 2020 111: 101488

8) ABSTRACT L11: Is the notion of “Pit1 acts as a helper of ETI” demonstrated? The presented data includes the HR conditioned by N. ben experiments using MHV/autoactive, but non-effector-triggered version. Is there any evident and literature backing up that Pit1 mediates effector-triggered immunity? Otherwise, this definitive sentence shall be reworded.

We have changed "Pit1 acts as a helper of ETI" to "Pit1 induces cell death" (page 4, line 11 in the revised manuscript with track changes).

9) L17: no need to add comma after “a new function”.

We have corrected this (page 4, line 17 in the revised manuscript with track changes).

10) In general, the gap of knowledge that this work addresses shall be much better articulated and novelty of this work shall be redefined. See the followings:

P5, L11: how duplicated NLR genes acquire new functions were rather very well studied in my opinion, e.g. L6/L7 and RPS4/RRS1 and its duplicate pair of RPS4B/RRS1B, alleles of NLRs causing autoimmunity as hybrid necrosis (Dangerous Mix genes). The reference 60 that they cited, for example, characterized allelic variation of the executor-sensor pair CHS3-CSA1 evolution. Authors shall clarify what had been known in the evolution of NLRs through duplication and sub/neo-functionalization.

P6, L8: Physical interaction between RPS4 and RRS1 as well as RGA4 and RGA5 were known, and there are other examples.

We have carefully reorganized the Introduction section to provide the gap of knowledge this work addresses and to redefine its novelty (page 6, line 17-page7, line 8 in the revised manuscript with track changes).

P6, L31-32: Compared to other NLRs, Pit1 downstream and signaling partners were identified with mode of action and localities. The sentence starting with “little is known” demotes their own finding. What authors meant would be that the mode of action of Pit1 NLR in relation to other NLRs are not known. Please articulate what is the main agenda of this research is here.

We have articulated the main agenda of this research in the Introduction (page 7, line 32 in the revised manuscript with track changes).

11) P7, L5: Unless identified, “the” ancestral Pit gene shall be referred to as “an” ancestral Pit gene.

We have corrected this (page 8, line 5 in the revised manuscript with track changes).

12) (related to major points) PAN3 results in suppressing Pit1 indicates its participation to Pit1-triggered immunity. Are there any sequence homology of PAN3 (and other PANs) to Pit1 and Pit2? Are there any reliable phylogeny to be presented to demonstrate their relationship as compared to other rice NLRs?

The amino acid sequence similarity between Pit1 and Pit2 is 88%, whereas that between Pit1 and PAN1-5 is no more than 34% (Extended Data Table 1). Additionally, our phylogenetic analysis reveals that except for Pit1 and Pit2, the other NLRs fall into distinct clades (Extended Data Fig. 2).

13) P11, L9-11: It appears to be an overstatement to interpret the disease resistance data of Pit2 KO presented in Figure1G. To further demonstrate Pit2 participating the Pit1-mediate resistance, double mutant analyses might be required. Rewording shall be made on this notion if further data cannot be provided.

We have corrected the overstatement (page 12, lines 3-4 in the revised manuscript with track changes).

14) P16: CH3-CSA1 > CHS3-CSA1

We have corrected this (page 17, line 31 in the revised manuscript with track changes).

15) NLR does not need to be italicized as the abbreviation itself represents protein domains. It is not the gene name, such as RPP1 or RPS2, of which abbreviation shall be italicized. "NLR (non-italics) genes" should be sufficient in delivering the messages of NLR-encoding genes.

We have corrected this throughout the manuscript.

REVIEWERS' COMMENTS

Reviewer #3 (Remarks to the Author):

The authors made substantial and thorough revision to differentiate their exciting studies to over the currently existing helper-sensor pair literature. I appreciate that the authors share the unpublished data on potential sensor function of Pit2. And they did an excellent job in setting a boundary of their work from the exciting helper-sensor function with their strong analysis on the evolutionary history of the duplication, diversification and maintenance of the important residues in both Pit1 and Pit2 in fine-tuning immunity. All these evolutionary genetics notions were supported by functional data with adequate explanation.

This is an excellent study explaining the evolutionary mechanism of how a duplicated paralog had evolved to regulate the other, likely being selected (through breeding) to presumably fine-tune the growth defense trade-offs. I cannot wait to read it again once published online. Congratulations on this huge achievement.

Please find my final suggestions for text edits, mainly on one paragraph in the introduction.

Page 6: Revise the section 17-22.

L17: This sentence is not correct. Plants fine-tune such gene functions to adapt “evolution” to a thing just does not sound right. Authors may consider how evolution occurs. This sentence may mean, “The function of plant NLRs in their breath of recognition spectra is believed to have evolved through the interaction with fast-evolving pathogens”.

L19-22: FLAX L consists of a single gene with numerous alleles of which Avr specificity spectrum had been extensively studied by Flor and other colleagues (see <https://www.ncbi.nlm.nih.gov/pmc/articles/PMC6640504/> for locus comparison of L to other M/N/P loci, and Ellis et al., PNAS, 1995). From this simple genetics, gene-for-gene hypothesis had been formulated. If the authors' intention is to highlight the evolution of NLR genes upon duplication and subfunctionalization, this example is not relevant.

The authors may focus their discussion in this paragraph on different allelic features of the paired NLRs.

L28 and Page6-7: the name “non-NLR ID” seems to indicate that they are non-NLR but ID. Either NLRs without ID or non-ID would be proper.

L29: seems to “have” acquired

Page17 L5: Pit1 and Pit2 genes exhibit (no s needed).

MS ID#: NCOMMS-20-16774

MS TITLE: An NLR paralog Pit2 generated from tandem duplication of *Pit1* fine-tunes Pit1 localization and function

We are grateful to the editor and Reviewer #3 for the critical comments and valuable suggestions that have helped us to improve our manuscript.

The revision has been made as follows according to the reviewers' comments.

REVIEWER COMMENTS

Reviewer #3:

1) The authors made substantial and thorough revision to differentiate their exciting studies to over the currently existing helper-sensor pair literature. I appreciate that the authors share the unpublished data on potential sensor function of Pit2. And they did an excellent job in setting a boundary of their work from the exciting helper-sensor function with their strong analysis on the evolutionary history of the duplication, diversification and maintenance of the important residues in both Pit1 and Pit2 in fine-tuning immunity. All these evolutionary genetics notions were supported by functional data with adequate explanation.

This is an excellent study explaining the evolutionary mechanism of how a duplicated paralog had evolved to regulate the other, likely being selected (through breeding) to presumably fine-tune the growth defense trade-offs. I cannot wait to read it again once published online. Congratulations on this huge achievement.

We are grateful for your positive comments regarding our manuscript.

Please find my final suggestions for text edits, mainly on one paragraph in the introduction.

2) Page 6: Revise the section 17-22.

L17: This sentence is not correct. Plants fine-tune such gene functions to adapt "evolution" to a thing just does not sound right. Authors may consider how evolution occurs. This sentence may mean, "The function of plant NLRs in their breath of recognition spectra is believed to have evolved through the interaction with fast-evolving pathogens".

Your phrasing is more suitable. We have revised that sentence to incorporate your suggested sentence (page 6, lines 1 and 2).

3) L19-22: FLAX L consists of a single gene with numerous alleles of which Avr specificity spectrum had been extensively studied by Flor and other colleagues (see <https://www.ncbi.nlm.nih.gov/pmc/articles/PMC6640504/> for locus comparison of L to other M/N/P loci, and Ellis et al., PNAS, 1995). From this simple genetics, gene-for-gene hypothesis had been formulated. If the authors' intention is to highlight the evolution of NLR genes upon duplication and subfunctionalization, this example is not relevant. The authors may focus their discussion in this paragraph on different allelic features of the paired NLRs.

As you rightly noted, the aim of this manuscript is to explore the distinct allelic features of the paired NLRs. Therefore, we have removed those sentences (page 6, line 4).

4) L28 and Page6-7: the name “non-NLR ID” seems to indicate that they are non-NLR but ID. Either NLRs without ID or non-ID would be proper.

We have corrected it (page 6, lines 22 and 23).

5) L29: seems to “have” acquired

We have corrected it (page 6, line 10).

6) Page 17 L5: Pit1 and Pit2 genes exhibit (no s needed).

We have corrected it (page 16, line 2).

Other changes)

7) To fit the Nature Comm format, we have edited the manuscript.

8) We have changed the reference paper and description on the selection signature on the sensor NLR RGA5 (page 16, lines 10-13).